# A safe, T cell-inducing heterologous vaccine against elephant endotheliotropic herpesvirus in a proof-of-concept study

Tanja Maehr [1,2,3] ✉, Javier Lopez[2], Gabby Drake[2], Frederico M. Ferreira[4], Richard Fraser[2], Rebecca Mckown[2], Reshma Kailath[5], Susan Morris[5], Adam Chambers[6], Leo P. Graves[6], Susan L. Walker[2], Akbar Dastjerdi [3], Katie L. Edwards [2], Helder I. Nakaya [7,8,9] & Falko Steinbach [1,3] ✉

We report the results of the world's first trial of a vaccine against elephant endotheliotropic herpesvirus (EEHV) in elephants. EEHV-induced haemorrhagic disease is a major threat to juvenile Asian elephants. A vaccine preventing severe disease and death would support conservation efforts for this endangered species. We developed a heterologous, recombinant modified vaccinia virus Ankara prime and adjuvanted protein boost vaccine, containing regulatory protein EE2 and major capsid protein. Vaccine design targeted Th1 and cytotoxic T cell responses, crucial for herpesvirus immunity. In a proof-of-concept trial, safety and immunogenicity were tested in adult elephants. A modified interferon-γ release (IFNG) point-of-care vaccine-specific whole blood assay was established to avoid sample transport-related loss of immune readouts and determine T cell responses by RT-qPCR first. Subsequently, RNA sequencing was utilised to investigate transcriptomic changes post-vaccination. No adverse reactions were observed following heterologous vaccination. *IFNG* responses to candidate antigens were detected against the pre-existing latent immunity in adult elephants. Over-representation analysis revealed induction of T cell-associated pathways. Thus, we show that the vaccine has a favourable safety profile and stimulates EEHV-specific T cell-biased immune responses, warranting further evaluation.

Elephant endotheliotropic herpesvirus haemorrhagic disease (EEHV-HD) is a significant cause of calf and juvenile mortality in both human-managed and free-ranging Asian elephants (*Elephas maximus*) and increasingly in African elephants (*Loxodonta africana*)[1,2]. Both elephant species are classified as "endangered" on the International Union for the Conservation of Nature's Red List[3,4]. Thus, the loss of young animals due to EEHV-HD poses a threat to the sustainability of conservation breeding programmes. Clinical disease and fatalities have predominantly been reported in Asian elephants and occur mainly between the ages of 1 and 8 years in this species[1].

[1]Department of Comparative Biomedical Sciences, School of Veterinary Medicine, Faculty of Health and Medical Sciences, University of Surrey, Guildford, United Kingdom. [2]North of England Zoological Society, Chester Zoo, Upton by Chester, United Kingdom. [3]Virology Department, Animal and Plant Health Agency (APHA) Weybridge, Addlestone, United Kingdom. [4]Cellular, Genetic, and Molecular Nephrology Laboratory (LIM-29), Hospital das Clínicas, University of São Paulo Medical School, São Paulo, Brazil. [5]Pandemic Sciences Institute, Nuffield Department of Medicine, University of Oxford, Oxford, United Kingdom. [6]Oxford Expression Technologies Ltd, Bioinnovation Hub, Oxford, United Kingdom. [7]Hospital Israelita Albert Einstein, São Paulo, Brazil. [8]Instituto Pasteur de São Paulo, São Paulo, Brazil. [9]Department of Clinical and Toxicological Analyses, School of Pharmaceutical Sciences, University of São Paulo, São Paulo, Brazil. ✉e-mail: tanja.maehr@apha.gov.uk; f.steinbach@surrey.ac.uk

Phylogenetically, EEHV-1 to -7 are most closely but distantly related to human cytomegalovirus (HCMV) and human herpesviruses (HHV)-6A, -6B and -7[5,6]. They are grouped into the *Betaherpesvirinae* as unique members of the *Proboscivirus* genus[7]. Asian elephants are susceptible to EEHV-1A, -1B, -4 and -5, which are endemic in this species[1]. The variants EEHV-1A and -1B are most associated with severe disease, with EEHV-1A accounting for most HD fatalities[8]. Most adult elephants are latently infected and act as reservoirs from which EEHV can reactivate and shed[1]. Waning of maternal antibodies correlates with EEHV-HD susceptibility in naïve calves suggesting that morbidity and mortality occur in animals experiencing primary infection rather than reactivation[9,10].

Whilst methods for early detection of viremia followed by intensive supportive therapy have improved[11,12], survival outcome in elephants remains uncertain. The absence of a viral culture and animal model system for EEHV and a limited toolbox for the elephant immune system relative to other animal species has hampered progress in antiviral drug and vaccine development. Accordingly, the development of an EEHV-1A vaccine that can prime the immune system of young Asian elephants sufficiently to prevent severe disease and death is a key research priority.

Since T cells play an essential role in controlling herpesvirus immunity, latency and reactivation[13–17], T cell-inducing vaccines may provide a path towards protection from EEHV-HD. For instance, cell-mediated immunity is the predominant mechanism by which replication of HCMV is controlled and severe HCMV disease predominantly occurs in patients with profound cellular immunodeficiency[18–20]. Inadequate cellular immune responses have also been linked with a higher risk of developing HHV-6-related disease[15,16,21,22]. Accordingly, a hallmark of infection with CMV and other herpesviruses is the continuous expansion and maintenance of a large pool of virus-specific, functional CD8+ effector memory T cells—a phenomenon termed "memory T cell inflation"[23,24] that correlates to CMV control and latency[25].

Given the above importance of T cells in herpesvirus immunity, this project aimed to devise a T cell-targeting vaccine that would elicit a strong Th1 and cytotoxic T cell response. Selecting T cell targets rather than neutralising B cell antigens also eliminates the possibility of antibody-dependent enhancement (ADE) through vaccination. While the risk of this phenomenon in EEHV infection is unknown, monocytes have been proposed as target and carrier cells for viral dissemination in EEHV-HD[26], thus implying that ADE might occur in EEHV infections.

Based on a genome comparison of the most immunogenic T cell antigens for HCMV, HHV-6, -7 and their EEHV orthologues[13–17,27], a literature review and a study that aimed to identify immunogenic EEHV T cell targets[28], the major capsid protein (MCP) and the EEHV-specific, putative regulatory protein EE2 were selected as EEHV vaccine candidate antigens. For antigen delivery, a modified vaccinia Ankara (MVA) virus and adjuvanted protein subunit approach in a heterologous prime-boost regimen were selected, as this has been proposed to elicit stronger, broader and/or longer-lasting immunity than homologous vaccinations[29]. Combining different vaccine modalities leverages their distinct immunological mechanisms to enhance the overall effect[30]. Viral vectors are particularly effective at priming T cell-mediated immunity because they mimic natural viral infection and efficiently present intracellular antigens to the host's immune system. In our vaccine design, this allowed us to exploit the capacity of MVA to robustly prime CD4+ and CD8+ T cells[31], a key requirement for providing immunologically naïve elephant calves with an early and effective cellular immune response against primary EEHV infection. Complementing this, the use of a subunit boost vaccine formulated with an adjuvant directs the immune response toward the expansion of Th1 responses[32], while using a distinct pathway of antigen delivery, processing and presentation. This strategy supports prolonged immune stimulation and facilitates the development of durable memory T cells.

MVA was considered a safe and effective vaccine platform for use in elephants since it is replication-deficient in mammalian cells and has thus been attributed a high safety profile[33,34]. MVA has the capacity to deliver multiple heterologous antigens and the ability to trigger both cellular and humoral immunity[35,36]. Importantly, MVA has been safely used for cowpox vaccination in Asian elephants in Europe before[37–39].

Protein subunit vaccines formulated in adjuvants also are well-established as safe and circumvent the need for whole virus in vaccine development[40]. Insect cell-baculovirus expression systems have been extensively used to produce veterinary and human viral subunit vaccines[41].

In this first in-elephant proof-of-concept trial, the primary aim was to evaluate the safety of this approach. To surmount the adverse effects of long blood sample transport times from the zoo to the laboratory on sample quality, we established a vaccine-specific interferon-γ (IFNG) release-based, point-of-care whole blood stimulation assay that can be carried out and preserved at the trial site for subsequent gene expression analysis, starting with measuring *IFNG* as a correlate of T cell activation. By subsequently sequencing total RNA, it was possible to utilise systems immunology approaches to deepen the analysis of the immunological responses to levels not explored in Asian elephants before.

## Results

The vaccination and sampling regimen is summarised in Fig. 1 to provide an overview of the study design and experimental setup.

### Vaccine safety and reactogenicity

Close monitoring of vaccinated elephants raised no safety concerns associated with the recombinant (r) MVA-EE2-MCP prime or the Montanide ISA 201 VG-adjuvanted subunit boost vaccine. No mild or moderate adverse events, for example symptoms such as local swelling, were observed at any point in any of the three elephants vaccinated during this study.

### Antigen-specific *IFNG* gene expression

*IFNG* as a strong correlate of Th1 immune responses was measured here by RT-qPCR gene expression analysis to assess cellular immune responses following whole blood stimulation with rEE2 and rMCP antigens before and after vaccination (Fig. 2).

### Latently infected elephants display a background immune reaction against EEHV antigens

*IFNG* mRNA was detected at significant levels in pre-vaccination blood samples stimulated with either the rEE2 or rMCP vaccine antigen. These recall responses before vaccination are supported by the notion that all three adult elephants were latently infected with EEHV as anticipated for adult animals, which is corroborated by historical EEHV shedding and serological data indicating humoral immunity to major viral glycoproteins[42].

Pre-vaccination blood samples from all three elephants displayed higher recall *IFNG* expression levels when stimulated with rEE2 than with rMCP. Interestingly, the baseline *IFNG* transcript levels correlated to the age of the animals, i.e., the oldest elephant had the highest baseline levels in response to both antigens (Fig. 2e, f).

### Vaccination with rEE2 and rMCP antigens induces memory recall responses

The first vaccinated elephant (A) received 20% of the eventual viral dose of MVA and half of the μg protein boost dose during the first rMVA-EE2-MCP prime and protein boost cycle to initially assess safety and determine the dose range, which was then increased in the second prime-boost cycle to the final dose used in the rest of the trial.

Nevertheless, memory recall responses were observed following the vaccination of elephant A: a 948- and 928-fold increase in *IFNG*

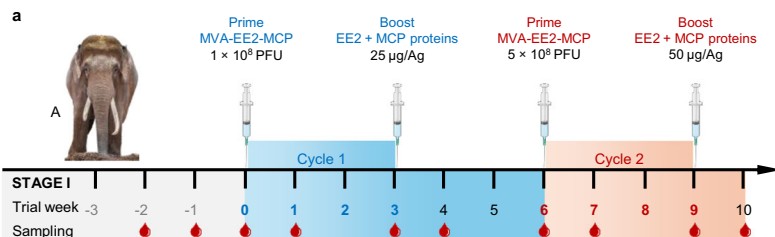

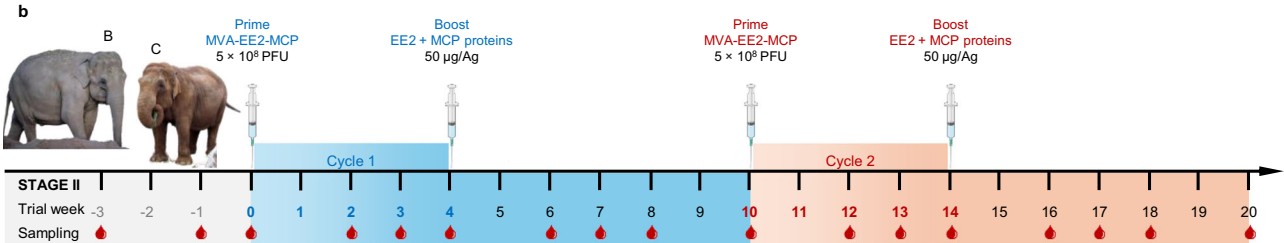

**Fig. 1 | Schedule of heterologous vaccination with elephant endotheliotropic herpesvirus (EEHV)-1A antigens EE2 and major capsid protein (MCP) and blood sampling strategies in the two-staged proof-of-concept and safety study.** Prime and boost vaccines were delivered via a recombinant (r) modified vaccinia virus Ankara (MVA) vector and in form of an adjuvanted water-in-oil-in water (W/O/W) protein subunit formulation, respectively. **a** In stage I (*n* = 1), elephant A was vaccinated with 1 × 10⁸ plaque-forming units (PFU) of rMVA-EE2-MCP and 25 µg of rEE2 and rMCP antigen (Ag) each in a first prime-boost cycle (cycle 1). In a second prime-boost course (cycle 2), the animal received 5 × 10⁸ PFU of the rMVA vaccine and 50 µg of each recombinant vaccine Ag in W/O/W. All vaccinations in stage I

were administered at 3-week intervals. **b** In stage II (*n* = 2), elephants B and C received two prime-boost courses of the higher doses tested for safety in cycle 2 of stage I and vaccinations were set further apart, with 4 weeks between prime and boost vaccinations and 6 weeks between cycle 1 and cycle 2. During both stages, blood and plasma samples were collected as indicated by the red drops below the timelines. Shaded areas indicate the three phases of the sampling and vaccination scheme: grey−pre-vaccination phase; blue−prime-boost cycle 1; red−prime-boost cycle 2. One element in this figure was created with BioRender.com. Elephant photos courtesy of Chester Zoo, used with permission. © Chester Zoo.

gene expression was detected in response to stimulation with the rEE2 and rMCP antigens in week (W) 3 after the first rMVA prime vaccination for this elephant. *IFNG* transcript levels in rMCP-stimulated blood also started to increase again after the first protein vaccination and peaked at a fold change (FC) of 1178 in W7. Boosting with a second protein injection did not enhance this response. *IFNG* expression was not boosted above baseline levels in response to the rEE2 antigen during the second vaccination cycle. This suggested that the 3-week intervals between each vaccination did not allow enough time for adequate contraction of *IFNG* expressing T cells (Fig. 2a, b).

Based on these results, intervals for stage II including elephants B and C were extended to 4 weeks between prime-boost vaccinations and 6 weeks between first and second vaccination cycles (Fig. 1). Both animals received the higher doses from the beginning, and the blood sampling schedule was changed to collecting blood for up to 3 weeks and 1 week prior to vaccination, on vaccination day, and 2, 3 and 4 weeks following each injection to better resolve the vaccine responses over time.

Although substantial variation between elephant B and C occurred regarding the magnitude of recall responses to rEE2 or rMCP antigens in post-vaccination samples, elevated *IFNG* expression or levels that exceeded baseline values after rMVA or protein boost vaccinations were observed in blood from both elephants after stimulation with rMCP (Fig. 2d, f). In blood obtained from elephant B after initial rMVA vaccination, *IFN*G recall responses induced upon stimulation with rMCP appeared to be boosted by the first protein vaccination with a 9340-fold increase of *IFNG* expression in W6 (Fig. 2d). While this was the maximum recall response to rMCP observed for this elephant and downregulation to almost baseline levels supervened, transcript levels were upregulated again and, during the second prime-boost cycle, reached a peak with a FC of 5658 two weeks after the second rMVA vaccination (W12). Although the final protein boost vaccination did not enhance *IFNG* gene expression in response to

rMCP stimulation, *IFNG* expression remained elevated and above baseline until the end of the study in this animal.

In blood from elephant C, *IFNG* gene expression was highly upregulated upon rMCP stimulation in W2 and W4 following the first vaccination with rMVA (FC W2: 33,627 and FC W4: 140,572 vs. FC W0: 428; Fig. 2f). While recall responses to this antigen did not exceed W4 levels at any of the following blood sampling dates, *IFNG* transcript levels were upregulated in response to the remaining vaccinations and appeared to be maintained above baseline following the final protein boost vaccination in this animal (Fig. 2f).

rEE2 stimulation of post-vaccination blood samples induced more modest memory recall responses in elephant B, where *IFNG* expression only exceeded the highest transcript levels measured in pre-vaccination samples following immunisation with the second rMVA vaccine in this animal (maximum FC: 6767, W14; Fig. 2c). In contrast, *IFNG* mRNA expression in blood from elephant C was upregulated beyond the highest baseline expression levels following stimulation after the first rMVA vaccination and reached a maximum FC of 108,873 4 weeks after the first protein boost vaccination (W8; Fig. 2e).

## Analysis of pre- and post-vaccination whole blood transcriptomes

As *IFNG* gene expression analysis yielded some encouraging results but retained ambiguity due to the pre-existing immunity to latent infection, we further investigated whole transcriptome changes in the two stage II elephants that had received the full vaccine doses at extended intervals.

### Post-vaccination transcriptomic changes in unstimulated blood samples

Our first interest was to identify changes in the transcriptomic profile of control (phosphate buffered saline (PBS)-treated) samples between

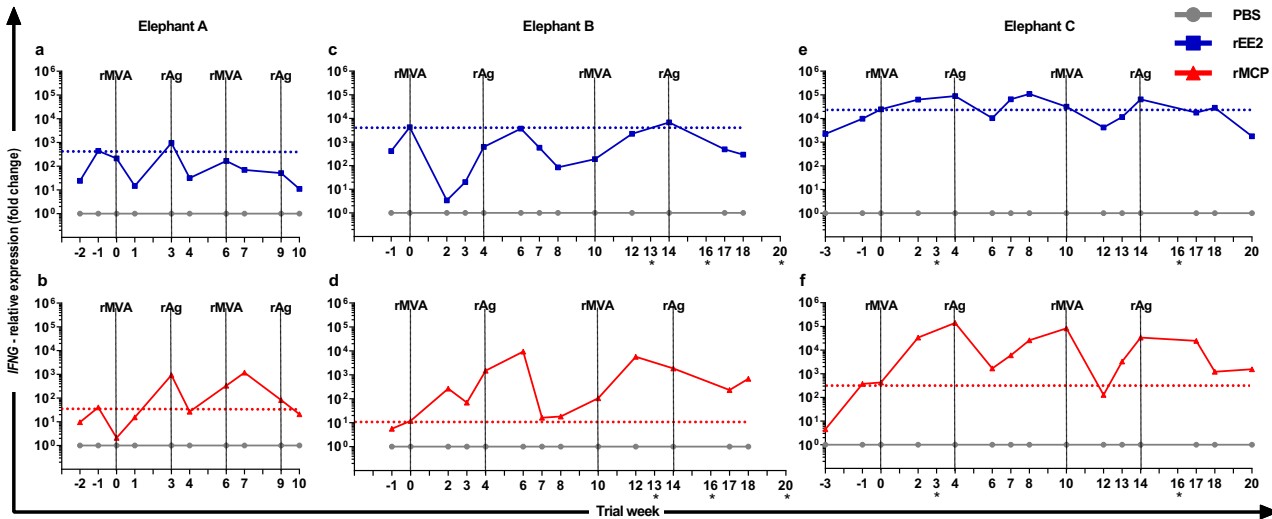

**Fig. 2 | Modulation of interferon-γ (*IFNG*) gene expression following stimulation of pre- and post-vaccination blood samples with recombinant (r) EE2 or major capsid protein (MCP) vaccine antigens (Ag).** The longitudinal *IFNG* gene expression levels in samples stimulated with rEE2 or rMCP relative to the expression in time-matched phosphate buffered saline (PBS)-treated controls over the stage I (**a**, **b**) and stage II (**c**–**f**) study periods are shown for the three participating elephants (*N* = 3). Data are presented for each individual animal (stage I: *n* = 1; stage II: *n* = 2). Vertical dashed lines indicate time of vaccination with either rMVA-EE2-MCP virus vector (rMVA) or adjuvanted rEE2-rMCP-subunit (rAg) formulation.

Horizontal dashed lines mark the highest *IFNG* expression level observed among baseline (pre-vaccination) samples. Through calculating the fold changes using the $2^{-(\Delta\Delta Ct)}$ method, *IFNG* mRNA levels were normalised to those of the reference gene elongation factor−1α and PBS controls were set as 1. Asterisks below the trial week numbers indicate that a blood sample could not be obtained from an animal at this time point (elephant C: weeks 3 and 16) or samples that were excluded from further analysis due to unsatisfactory downstream quality checks (elephant B: weeks 13, 16 and 20). Source data are provided as a Source Data file.

pre- and post-vaccination time points (contrast A, Supplementary Fig. 1 and Supplementary Data 1a). This revealed that the expression of a total of 2678 genes was significantly modulated after vaccination. Of these differentially expressed genes (DEG), 944 were upregulated and 1734 downregulated. FC of upregulated DEG ranged from 1.3 to 15.6 and from −1.3 to −60.8 in downregulated genes (adjusted *p* value (padj) ≤0.05).

### CD8+ and CD4+ T cells are upregulated following vaccination

To further elucidate the mechanism of action induced by vaccination, we looked for enriched cell signatures associated with the DEG post-vaccination in unstimulated blood. Using the significantly upregulated DEG, over-representation analysis (ORA) identified an association with gene sets specific for CD8+ T cells (*p* <0.001) and CD4+ T cells (*p* = 0.003) in the Human Gene Atlas (Fig. 3). The DEG overlapped with 50/602 genes in the Human Gene Atlas *CD8+ T cells* gene set. Twenty-nine of these genes were shared with the *CD4+ T cells* gene list, while another 11 genes were specific for *CD4+ T cells*, thus mapping a total of 40/533 to these.

Besides the association with the T cell gene sets above, a number of obvious immune mediators, such as interleukin 15 (*IL15*), *CD28* and the inducible T cell costimulator *ICOS*, were identified among the upregulated genes post-vaccination. Of note, the second most upregulated gene with an FC of 11.5 in the entire gene set was a LOC gene (LOC126061467) predicted to translate into a granzyme A (GZMA)-like protein (Supplementary Data 1a), granzyme A being involved in cytotoxic T lymphocyte-induced apoptosis.

### CD14+ monocytes are downregulated following vaccination

Many immune-relevant DEG were significantly downregulated following vaccination, with the most prominent immune genes being a *CCL8-like* gene (LOC126062469; FC: −60.8), the chemokines *CXCL11, CXCL10, CCL24, CXCL6, CXCL1* (FC: −48.5, −23.7, −20.3, −15.4 and −9.4, respectively), the interferon-stimulated gene *ISG15* (FC: −11.4), the pro-inflammatory cytokine *IL1B* (FC: −8.7) and the interferon-regulatory transcription factor *IRF7* (FC: −7.6) (Supplementary Data 1a).

A high number of these DEG were associated with the innate immune system and the downregulation of innate immunity was corroborated by ORA of the DEG using the Human Gene Atlas. Here, particularly the *CD14+ Monocyte* gene set and a gene set relating to B cells (*721 B lymphoblasts*) were significantly downregulated (Fig. 3); and so were the *CD56+ Natural killer (NK) cell* gene set and a plasmacytoid dendritic cell-related gene set (*BDCA4+ Dendritic cells*), albeit to a lesser extent.

### Gene expression in response to rEE2 and rMCP stimulation in vitro before and after vaccination

To further differentiate vaccine-induced responses from the latent EEHV immunity background of our study animals, contrast B (the pre-vaccination differential whole blood gene expression following rEE2 and rMCP antigen stimulation relative to time-matched PBS controls) was assessed first. Subsequently, these transcriptome changes could be compared to those in contrast C, which was generated to reveal the gene expression changes following antigen stimulation post-vaccination (Supplementary Fig. 1). In contrast B, 2957 DEG (padj ≤0.01) were present; a number that was substantially enhanced in contrast C to 6686 DEG (padj ≤0.01) (Supplementary Data 1b and 1c). There was an overlap of 2497 DEG between the pre-vaccination and post-vaccination dataset, of which 1341 genes were upregulated and 1155 genes downregulated. The number of upregulated genes in contrast B and C was 1489 and 3440, respectively.

The transcript showing the highest upregulation after antigen stimulation pre-vaccination was *CXCL13* (FC: 3554), encoding a chemokine which preferentially attracts B lymphocytes and has been proposed as a plasma biomarker of germinal centre activity[43]. A *LIF-like* gene (LOC126065436), *IFNG* and a *IL36G-like* gene (LOC126060847) ranked 5th to 7th among the top upregulated genes pre-vaccination (FC: 577, 524 and 512), respectively. Similarly, these genes were also among the top upregulated genes post-vaccination, where the *IL36G-like* gene was the top induced gene (FC: 6573) while *IL1A* was the 2nd most upregulated DEG (FC: 4511, rank 30 in contrast B) and *CXCL13* was the 5th top upregulated gene (FC: 2517). *IFNG* represented the 4th top

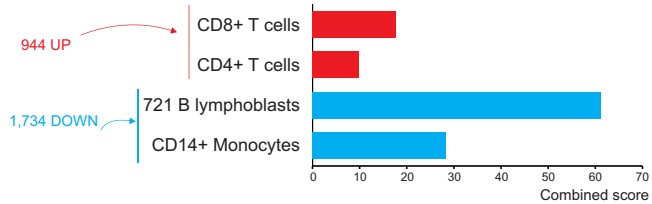

**Fig. 3 | Over-representation analysis of differentially expressed genes (DEG) in unstimulated whole blood samples post-vaccination (contrast A).** Over-representation analysis was performed using DEG identified from samples of two elephants ($n = 2$). A total of 944 upregulated and 1734 downregulated DEG (adjusted $p$ value ≤0.05) were analysed using the Human Gene Atlas database. The red and blue bars represent the combined scores of gene sets enriched for upregulated and downregulated cell types, respectively. In unstimulated blood samples, CD8+ and CD4+ T cells were significantly over-represented, while monocyte and B cell signatures were downregulated following vaccination. Significantly enriched pathways were identified using a one-sided Fisher's exact test (H₀: no association), with correction for multiple comparisons by applying the Benjamini-Hochberg method, and ranked by the combined score (z-score of the observed and expected means of the DEG and pathways overlap). Pathways with a $p$ value ≤0.05 were considered significant.

upregulated gene among the DEG in the post-vaccination dataset and was also induced with a higher FC (2665) than in contrast B. This demonstrated that *IFNG* was a good choice to determine antigen-specific reactions by RT-qPCR but also that the interpretation of single genes is particularly difficult in an immune memory situation.

Also among the top 25 induced genes following antigen-stimulation in both pre-and post-vaccination blood were *IDO2* (FC B: 375; FC C: 1168) and *IL1B* (FC B:110; FC C: 980). Additional shared DEG present in the top 25 upregulated transcripts post-vaccination but ranking below the top 87 induced genes pre-vaccination included *CCL22* (FC B: 36; FC C: 1221), *CCL24* (FC B: 7; FC C: 431) and a *GROG-like* gene (*CXCL3-like*; LOC126077077; FC B: 24; FC C: 427).

In addition to these top ranked immune genes, a substantial number of other immune-relevant genes was detected among the remaining upregulated DEG in both pre- and post-vaccination tran-scriptomes. There was a considerable overlap between the chemo- and cytokine (receptor) signature induced during recall responses before and after vaccination, involving chemokine (receptors) *CCL5, CCL20, CCL22, CCL24, CX3CL1, CXCL1, CXCL6, CXCL8, CXCL13, CCR7, CXCR2, CXCR4* and cytokine (receptors) *CSF2, CSF3, IFNG, IL1A, IL1B, IL6, IL10, IL12A, IL12B, IL18, IL23A, IL27, IFNGR2, IFNLR1, IL1R1, IL2RA, IL4R, IL12RB2, IL21R, TNFSF13B, TNFSF14, TNFRSF4, TNFRSF9, TNFRSF10B, TNFRSF18, TNFRSF25, XCL1* and two LOC genes annotated as *IL22* (LOC126075761) and *IL22-like* genes (LOC126075763).

Both datasets also shared interferon-stimulated genes *ISG15* and *ISG20* and several transcripts for cell surface proteins of immune genes including *CD40, CD48, CD80, CD83, CD84, CD86, CD93, CD163, CD274 (PDL1), CD278 (ICOS), CD317 (BST2; tetherin)* and immune-associated transcription factors and modulators such as *AHR, ARID5B, ATF3, BACH2, BATF, BATF3, BHLHE40, ELF1, ETV3, IRF4, IRF7, JUNB, KLF4, KLF5, KLF6, NFIL3, NFKB1, NFKBIA, NFKBIE, NFKBIZ, PCNA, RBPJL, REL, RELB, RORA, RUNX1, RUNX3, SMAD1, SMAD7, SOX4, STAT4, ZBTB10, ZBTB18, ZBTB21* and *ZBTB46*.

Of note, the FC of the immune relevant DEG shared pre- and post-vaccination were usually higher or at a similar level post-vaccination, with few exceptions such as *CXCL13* (the top upregulated gene before vaccination; FC B: 3554; FC C: 2517), complement component *C8G* (FC B: 50; FC C: 20), *CX3CL1* (FC B: 34; FC C: 8), which can contribute to the recruitment of effector T cells to peripheral tissues and lymphoid organs[44] and participates in the adhesion between monocytes and endothelial cells[45], and *IL12A* (FC B:16; FC C: 11), a subunit of IL12 that plays an important role in the activities of T lymphocytes and NK cells.

Only a few genes with known or proposed immune functions were uniquely expressed before but not after vaccination. Of these, the decoy receptor *IL1R2* (FC: 42) and a LOC126068746 predicted to encode a TLR13-like protein (FC: 22) ranked in the top 150 upregulated DEG. Further amongst the DEG that were solely upregulated pre-vaccination were the antiviral cytokine *IFNB1*, pro-inflammatory *IL16* (functions as a chemoattractant and modulator of T cell activation[46]), *TNFRSF8* (expressed by activated T and B cells and has been shown to limit the proliferative potential of CD8+ effector T cells[47]), the danger receptor *TLR7* and a LOC gene (LOC126079100) predicted to code for a TRIM21-like protein (whereby TRIM21 is a putative modulator of IRF3 stability and a positive regulator of strength and duration of primary antiviral responses[48]).

In contrast, a large set of 2099 unique DEG, among them a ple-thora of immune-related transcripts was upregulated after vaccina-tion. These included, within the top 25 induced genes, *IDO1* (FC: 1083), *CXCL9* (FC: 484) and a *CXCL8-like* transcript (LOC126062469, FC: 447). Further immune genes detected within the top 150 DEG were *MMP10* (FC: 314), *CXCL11* (FC: 276), *IL21* (FC: 238), *CCL17* (FC: 171), *CCL1* (FC: 122), a *GAL10-like* gene (LOC126085583; FC: 116), *CLEC12B* (FC: 90), *IL17A* (FC: 85), *IL2* (FC: 76) and *ICAM5* (FC: 70). Other uniquely present transcripts from different immune gene families included cyto- and chemokine (receptors) *CXCL9, CXCL10, CXCL16, IL2, IL4, IL7, IL13, CCR4, CCR6, IL1RL1, IL2RG, IL10RB, IL12RB1, IL15RA, IL17A, IL18R1, IL20RB, IL27RA, TNF, TNFSF4, TNFSF10, TNFRSF1B, TNFRSF14*, the CD markers and ligands *CD1D, CD1E, CD2AP, CD4, CD28, CD38, CD40LG, CD44, CD53, CD68, CD70, CD74, CD109, CD226, CD247* and immune-relevant transcription factors and modulators such as *ATF4, ATF5, ATF6, BBX, CITED2, CREB5, E2F3, E2F7, ELF2, EOMES, ETV6, ETV7, FOXO1, FOXP4, GATA3, GATA6, GFI1, JUN, KLF12, MAF, MAX, NFKB2, RBPJ, RELA, STAT1, STAT2, STAT3, STAT5B, TBX21 (TBET)*, a *ZAP70-like* gene (LOC126060770) and *ZBTB1*.

This vast expansion of uniquely upregulated DEG and the increa-ses of FC of shared DEG seen post-vaccination demonstrated that a biologically meaningful interpretation of single genes was almost impossible and biased. Hence, we conducted an over-representation pathway analysis on antigen-specific upregulated genes pre- and post-vaccination to better understand the processes involved in latency protection and those stimulated or induced by this vaccine.

## Over-representation analysis (ORA) using the gene ontology biological process (GOBP) database

Using the Gene Ontology Biological Process (GOBP) database first, ORA of significantly upregulated DEG in response to antigen stimula-tion before and after vaccination revealed that both conditions were enriched for a large amount of Gene Ontology (GO) terms with padj ≤0.05 and shared many of these terms (Supplementary Data 2a, b). More so, a large number of these GO terms was redundant with reference to the same processes, cell types or genes. Accordingly, we carried out a redundancy reduction approach to reduce the number of terms ultimately referring to the same set of genes (Supplementary Data 3a, b). The resulting lists of GO terms were analysed for those occurring pre-vaccination only, both pre- and post-vaccination or post-vaccination only (Supplementary Data 3c, d). Focussing on pathways directly linked to the immune system (Supplementary Data 3c), it became evident that pathways linked to an innate (NK cells) or early, inflammatory immune reaction (IL17) seemed to peak pre-vaccination (Fig. 4). A considerable number of GO terms, including antiviral pathways and those linked to the adaptive immune system (with emphasis on IFNG and IL12) were significantly enriched in both pre- and post-vaccination contrasts but gene overlaps were higher after vaccination. Additional terms referring to Th1 cells and unique terms relating to IL15 were present post-vaccination. Furthermore, two sets of GO terms were relatively novel upon vaccination: one set referring to the activation of B cells, the other to the innate immune system,

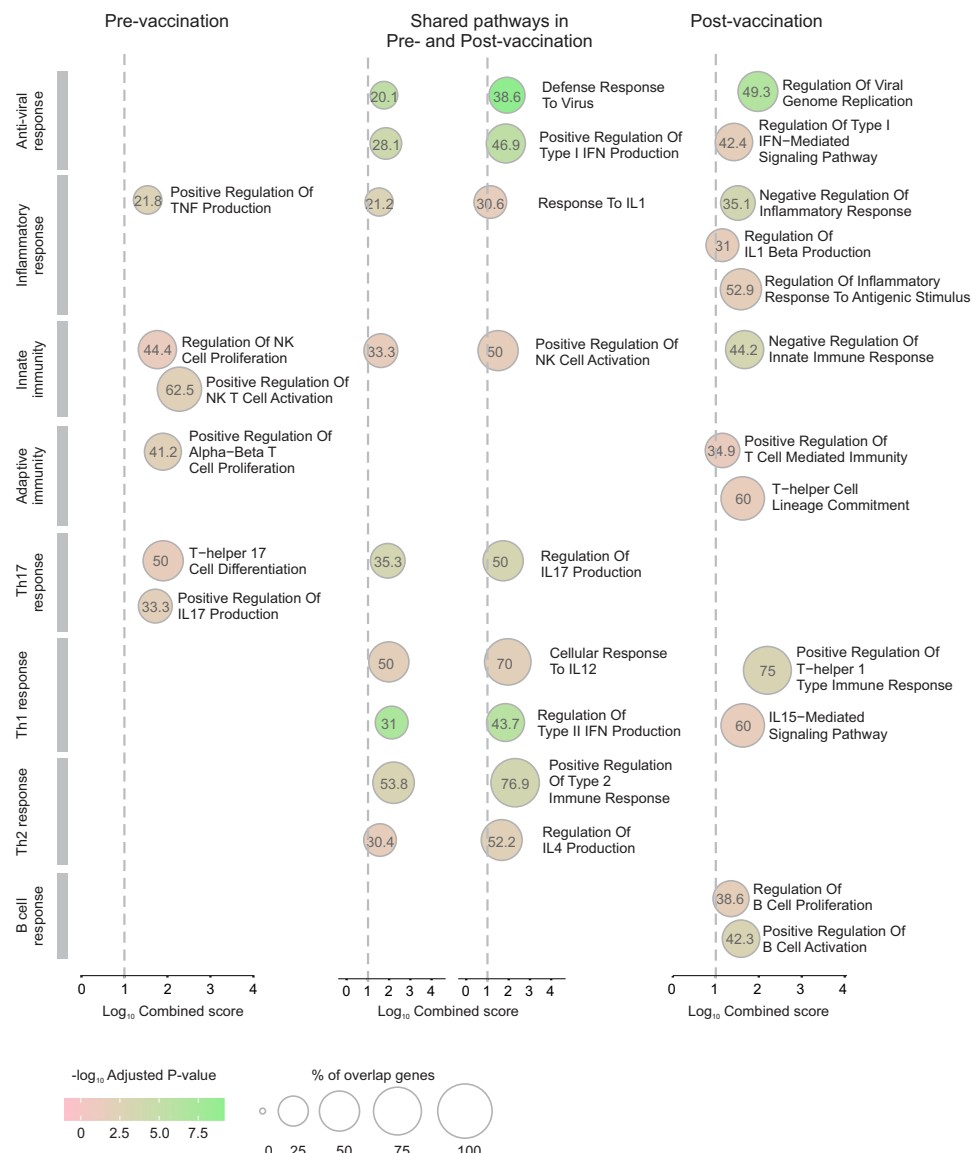

**Fig. 4 | Gene Ontology (GO) over-representation analysis of differentially expressed genes (DEG) upregulated upon antigen stimulation before and after vaccination.** Over-representation analysis was performed using DEG identified from samples of two elephants ($n = 2$). Representative immune-relevant GO terms following reduction of redundant terms (>85% gene overlap) in contrasts comparing unstimulated vs. antigen-stimulated samples pre- and post-vaccination are shown according to their occurrence in the pre-vaccination comparison only, in both pre- and post-vaccination comparisons and post-vaccination comparisons only. DEG with an adjusted $p$ value (padj) ≤0.01 were analysed. Significantly over-represented GO terms were identified using a one-sided Fisher's exact test ($H_0$: no association), corrected for multiple comparisons with the Benjamini-Hochberg method, and ranked according to the combined score (z-score of the observed and expected means of the DEG and pathways overlap). The cut-off for consideration of GO terms as significant was padj ≤0.05. IFN interferon, IL interleukin, NK natural killer, Th T helper, TNF tumour necrosis factor.

particularly concerning the regulation of inflammatory responses and the production of IL1B (Fig. 4).

## ORA using the Reactome database

With GOBP providing a large, and in parts not very precise output of pathways, we also used Reactome as another database with immunological terms to review the immune activation in stimulated blood before and after vaccination. Here again the number of pathways identified increased significantly after vaccination (Supplementary Data 4a, b). For further consideration we used a cut-off of padj ≤0.01 and a combined score ≥50 pre-vaccination and ≥100 post-vaccination to focus on top regulated immunologically relevant pathways only.

In both contrasts, pathways relating to IFNG, but also to IL4/IL13 and IL12 were over-represented, demonstrating an adaptive immune response. Interestingly, also IL10 and IL35 pathways could be seen in common pointing to limiting immune responses, thus demonstrating a complete immune response both limiting the latent EEHV and extended upon vaccination.

In the post-vaccination contrast, a number of additional cytokine-related pathways were identified, specifically, IL18, IL21, IL23 and IL27. Also, we identified an enrichment of innate immunity-linked, specifically TNFR- and IL1-related pathways in Reactome too (Table 1).

## Discussion

EEHV can cause a devastating disease in juvenile Asian elephants and in the absence of efficacious antiviral therapies mortality remains high. We report the results of the world-first EEHV vaccination trial in elephants, which was designed to demonstrate the in-elephant safety and proof-of-concept for the use of a heterologous, dual antigen vaccine against EEHV-1A.

**Table 1 | Over-representation analysis of differentially expressed genes (DEG) upregulated upon antigen stimulation before and after vaccination using the Reactome pathway database**

| Over-represented pathways before vaccination | Combined score | Over-represented pathways after vaccination | Combined score |
|---|---|---|---|
| | | Interleukin-21 Signaling | 595.79 |
| | | Interleukin-1 Processing | 465.40 |
| Interleukin-10 Signaling | 428.00 | | |
| | | Interleukin-35 Signaling | 332.19 |
| | | Interleukin-10 Signaling | 281.92 |
| Regulation of IFNG Signaling | 219.89 | | |
| | | Interleukin-1 Signaling | 169.22 |
| | | Interleukin-1 Family Signaling | 163.20 |
| | | Interleukin-23 Signaling | 153.09 |
| | | OAS Antiviral Response | 153.09 |
| Interleukin-4 and Interleukin-13 Signaling | 141.08 | | |
| OAS Antiviral Response | 131.14 | Regulation of IFNG Signaling | 137.34 |
| | | Interleukin-12 Family Signaling | 135.54 |
| | | Interleukin-4 and Interleukin-13 Signaling | 127.69 |
| | | Interleukin-27 Signaling | 121.94 |
| | | DDX58/IFIH1-mediated Induction of Interferon-Alpha/Beta | 115.30 |
| Interleukin-35 Signaling | 114.29 | Signaling by CSF3 (G-CSF) | 112.89 |
| | | TNFR1-induced NFkappaB Signaling Pathway | 111.83 |
| | | Interleukin-18 Signaling | 109.23 |
| | | Regulation of TNFR1 Signaling | 103.27 |
| Signaling by CSF3 (G-CSF) | 87.79 | | |
| Antiviral Mechanism by IFN-stimulated Genes | 61.83 | | |
| Interferon Gamma Signaling | 60.72 | | |
| Interleukin−12 Family Signaling | 57.94 | | |

Representative immune-relevant pathways in contrasts comparing unstimulated vs. antigen-stimulated samples pre- and post-vaccination are listed ranked by their combined scores. DEG with an adjusted p value (padj) ≤0.01 were included in the analysis. Over-representation was assessed using a one-sided Fisher's exact test, with Benjamini-Hochberg correction for multiple comparisons. The combined score is defined as the z-score of the observed and expected means of the DEG and pathways overlap. Only pathways with padj ≤0.01 and a combined score ≥50 (pre-vaccination) and ≥100 (post-vaccination) were deemed significant.
Pathways that were over-represented in both pre- and post-vaccination comparisons are presented in italics.

Analogous to phase I proof-of-concept vaccination trials, adult animals were used to monitor for reactogenicity and adverse effects of the rMVA-EE2-MCP prime and the Montanide-adjuvanted rEE2-rMCP-subunit boost during two prime-boost courses. The complete absence of local or systemic reactions or any other adverse events in all three adult elephants suggests that both rMVA and rEE2 and rMCP subunit antigens as well as the ISA 201 VG adjuvant are safe and very well tolerated in Asian elephants. The safety of rMVA was in line with the application of MVA as a vaccine to protect elephants from cowpox-induced disease[37,38]. The use of ISA 201 VG as adjuvant was considered a safe choice in the context of a T cell-targeting vaccination strategy since it enhanced cellular immune responses in vaccination protocols for pathogens such as influenza and foot-and-mouth disease virus in livestock species[49–51]. While there are no regulations regarding the purity of antigens in vaccines, we considered it warranted to start with a high purity (≥90%) of recombinant antigen to reduce the likelihood of adverse effects of the subunit vaccine[52,53].

The analysis of cellular immune responses focussed initially on *IFNG* gene expression as a well-established marker for such. That all three elephants had pre-existing immunity due to their latent infections was not necessarily surprising since all adult elephants may be considered latent EEHV carriers[54] and our study cohort was previously exposed to multiple EEHVs, including the target EEHV-1A[12,42]. Notably, low levels of antibodies against both EE2 and MCP were detectable in all elephants prior to vaccination (Supplementary Fig. 2), which corroborates latent infection.

Despite this latency-related immunity background, *IFNG* expression levels that exceeded baseline values in response to rMCP stimulation were observed in most post-vaccination samples, particularly in those from the two animals that had received the higher vaccine doses at extended intervals in stage II, indicating a vaccine-induced enhancement of recall responses to this antigen. rEE2 induced more modest *IFNG* memory recall responses. Both proteins were shown to be immunodominant in a study investigating T cell responses to nine EEHV proteins, where EE2 elicited broad IFNG responses in peptide-stimulated PBMC[28]. It had been proposed that EE2 as a putative regulatory protein might be a stronger target for T cells during latent infection. These findings may be reconciled with our results in that rMCP elicited recall responses from a lower background and then of relatively higher magnitude compared to rEE2 that was already a more prominent target of circulating T cells in latent infection.

The magnitude of response to the two vaccine antigens in both the highest baseline and post-vaccination samples was markedly higher in the oldest animal, elephant C. While individual variation of vaccine responses is common, herpesviruses are also known to induce T cell memory inflation over time[55], due to the continued maintenance and expansion of antigen-specific T cell populations[56,57]. CMV-specific T cell memory inflation depends on the presentation of viral antigens throughout the latent phase of infection[58]. In line with our data, it may be hypothesised that the higher *IFNG* recall responses are related to this phenomenon.

Against the backdrop of the active immunity in latent infection, gene expression analysis using *IFNG* as a singular readout for T cell activation had limitations. Therefore, we complemented the analysis by making use of systems immunology approaches which allowed an unbiased, in-depth analysis of gene expression and cellular pathways. So far only a very limited number of studies using transcriptomics analyses for such purpose exist in wildlife species[59–62].

Indeed, the comparison of unstimulated pre- and post-vaccination blood transcriptomes demonstrated the modulation of a substantial number of genes in response to vaccination. ORA of these DEG indicated a significant enrichment of CD8+ and CD4+ T cell genes supporting that T cells were specifically triggered by this vaccine approach. Furthermore, the suppression of B cells in unstimulated samples also demonstrated a modulation to focus the immune response to vaccination on T cells rather than antibody production. Hence, transcriptome analysis of unstimulated blood samples overall indicated that the heterologous vaccine is effectively inducing a T cell response while modulating the activity of innate immune cells and B cells.

We further analysed and compared antigen-stimulated (memory) responses before and after vaccination to better discriminate the vaccine-induced immune responses from a latency-induced immunity. Moreover, such an analysis can provide information on the immune reaction that could be activated in vivo upon re-encounter with antigen, and similar analyses in future vaccination studies may also provide potential markers of protection.

The absolute number of statistically highly significant DEG induced by antigen stimulation before and after vaccination exceeded our expectations. That vaccination with our heterologous prime-boost vaccines instigated an enhanced recall response is evident from the 3.3 times greater number of DEG induced post-vaccination. However, most of the shared genes which were upregulated upon antigen stimulation before vaccination were also upregulated after vaccination. The higher fold induction observed following vaccination demonstrated a vaccine-induced boost effect on a shared recall gene signature, while the uniquely expressed genes indicated a broadened and, potentially, qualitatively different memory response post vaccination.

This was reflected in ORA which revealed that a range of pathways were commonly induced both before and after vaccination; however, fewer pathways were uniquely induced in response to antigen challenge in latent infections before vaccination as opposed to a substantial number of pathways uniquely upregulated after post-vaccination antigen-stimulation.

Recall responses of latently infected unvaccinated elephants were more biased towards NK, innate and pro-inflammatory Th17 adaptive pathways. The overlapping immune reactions pre- and post-vaccination show an emphasis on anti-viral pathways, but also Th1 and some Th2 reactions. Notably, pathways relating to IL10 and IL35 were also over-expressed in response to antigen stimulation before and after vaccination, demonstrating that strong memory responses also contain regulatory feedback mechanisms for their limitation.

Further to enhanced antiviral responses, additional Th1-related terms were present in post-vaccination ORA, demonstrating a role for IL15, IL18, IL21, IL23 and IL27 pathways. All of these cytokine signalling pathways are integral to the positive regulation of T cell responses: IL15, IL18 and IL21 have significant roles in the activation of T cell functions and can act in synergy to enhance IFNG production in these cells[63]. IL15 is vital for the homoeostasis of CD8+ memory T cells[64], and IL21 enhances the cytotoxic activity of CD8+ T cells during acute and chronic viral infection[65]. IL21 and IL18 are promotors of the differentiation and activation of CD4+ T cells[65,66]. IL18 increases IFNG production to amplify the Th1 response in the presence of IL12[66], for which enriched pathways are present in pre- and post-vaccination ORA. Furthermore, IL23 can promote survival and proliferation of CD4+ memory T cells[67,68], and IL27 can promote the differentiation of naïve T cells into Th1 cells and exhibits anti-inflammatory properties by inducing the expression of IL10 and limiting the differentiation of Th17 cells[69].

Intriguingly, antigen stimulation after vaccination also induced a prominent upregulation of genes enriched in pathways referring to inflammatory responses such as IL1- and TNFR1-related pathways. *IL1A* and *IL1B* were also the second and 14th most upregulated DEG, respectively. While this seems counter-intuitive to the other results, it must be noted that this was in response to antigen stimulation and points towards a trained immunity response that would be beneficial, particularly for younger, naïve animals, as a result of the vaccination.

Despite our vaccine being primarily designed to induce a T cell response, western blot analysis demonstrated that the vaccine elicited measurable humoral responses, with antibody levels increasing post-vaccination compared to baseline (Supplementary Fig. 2). A degree of humoral activation against the vaccine antigens was expected and the over-represented pathways fit well with the activation of Th cells providing help to B cells, as well as some pathways pointing directly towards a secondary activation of B cells. Although antibodies induced by our vaccine would be of limited use in the immune defence, ancillary serological data may in the future serve as a diagnostic marker of vaccination.

While our vaccine targets effector T cell immune responses, whereby the Th cells might indirectly support a broad range of B cells upon infection and thus also promote a more rapid increase in neutralising antibodies, more direct approaches targeting EEHV-specific glycoproteins, particularly gB and the gH/gL complex, to elicit neutralising antibody responses have been undertaken in mice. The first two such studies tested the induction of gB-specific responses with rMVA and a protein subunit approach separately[70,71]. The most recent study uses a multivalent glycoprotein mRNA vaccine that demonstrated potential in a murine model[72]. Both neutralising antibody- and T cell-based strategies are valid avenues for EEHV vaccine development, and future vaccine development or optimisation efforts may benefit from integrating both strategies to enhance the breadth of immune responses and potentially extend cross-protection across EEHV genotypes. Our vaccine design specifically targeted EEHV-1A, the genotype most associated with fatal haemorrhagic disease in Asian elephants. The amino acid sequence identity with EEHV-1B is >98% for EE2 and MCP (Supplementary Fig. 3), suggesting that the immune responses elicited would extend cross-protective effects to this genotype. While MCP retains relatively high sequence homology with other EEHV types affecting Asian elephants, such as EEHV-5 and EEHV-4, the percent identity for EE2 is substantially lower, reducing the likelihood of meaningful cross-protection against these genotypes.

Because the elephant immune system differs remarkably from that of standard pre-clinical species (and other non-Afrotherian mammals)[73,74], classical models like mice or pigs would fail to accurately predict vaccine performance. More so, all principles of the vaccine (recombinant MVA to carry viral antigens and subunit vaccines) have been extensively studied in model systems. Therefore, in line with 3 R animal experimental principles, we chose to conduct our proof-of-concept study directly in elephants since an in-target-species trial in a real-world setting provides an accurate reflection of the vaccine's safety and immunogenicity, avoiding reliance on models that poorly recapitulate elephant immunology.

While our study interprets post-vaccination immune signatures as EEHV antigen-specific responses elicited by the heterologous prime-boost regimen, it is important to acknowledge that we cannot delineate the contribution of the different vaccine components employed during the repeated heterologous prime-boost regime. Some effects may be attributable to the rMVA or other vaccine components, rather than exclusively to the vaccine antigens. For instance, it is possible that the rMVA through its MVA backbone has re-activated already existing, latency-related T cell responses that underlie the results in contrast A where no antigen stimulation was applied. Similarly, we cannot draw a conclusion to which extent the MVA vector, as a live-attenuated virus, drove the putative trained immunity responses we see in the post vaccination ORA. The most prominent responses observed pre- and post-vaccination, however, were antigen-specific in contrasts B and D.

Taken together, these results demonstrate that the vaccine induced an overall antigen-specific T cell-driven immune response as

intended through our selection of T cell vaccine antigens and vaccine platforms. Our transcriptome analyses of vaccine responses show that systems immunology approaches can be successfully applied in species beyond livestock or companion animals in order to overcome limitations of single-parameter readouts or a lack of immunological tools as is often the case for non-model organisms. We used stringent statistical criteria for the RNA sequencing (RNA-seq) analysis; however, our data interpretation was limited in that we could only study two elephants. Nevertheless, this approach will be informative for the identification of immune correlates in future studies, not only for further studying naïve calves and EEHV, but also other vaccine studies in non-model animals as our results have far exceeded expectations.

Thus, this pilot study has laid a foundation for the successful implementation of immunogenicity and efficacy vaccination trials in larger cohorts in Asian elephant range countries and may provide a roadmap for vaccination studies of other wildlife diseases. Further to safety and immunogenicity, production, transport and storage considerations will also have to be taken into account for successful implementation of vaccines in range countries. While our vaccine approach was based on best established scientific practice for safety and immunogenicity in large animals, including elephants, the platforms used are also suitable for field conditions. MVA vaccines and ISA 201 VG-adjuvanted vaccines may, at least for short periods, be stored and/or transported at 2 to 8 °C, thus eliminating the need for ultra-cold chain transport and storage. One drawback for the choice of rMVA is the fact that it constitutes a GMO which, despite its safety profile, may limit its use in some countries. MVA is a highly attenuated and replication-deficient vector classified as a Risk Group 1 agent, indicating minimal risk to both human health and the environment[33,34]. Although shedding data are rarely reported in clinical studies, available evidence suggests that MVA persists in the body for less than 48 hours, and detection of shed material in excreta is short-lived, consistent with rapid viral clearance due to MVA's replication-deficient properties, which mitigates concerns about environmental release or transmission to untreated individuals[75].

The implementation of a small-volume, whole blood stimulation assay for the assessment of vaccine antigen-specific responses that can be set up at the trial site using minimal equipment, followed by sample preservation in RNA stabilisers and storage or transfer to the investigating laboratory for downstream processing, facilitates immunogenicity testing in clinical and field trials.

In conclusion, the results of this first-in-elephant proof-of-concept EEHV vaccine trial demonstrate that such heterologous prime boost vaccines can be safely used in this species. The systems immunology approaches applied in this species for the first time demonstrated a vaccine-based, antigen-specific Th1 and cytotoxic T cell-skewed immune response. These vaccine responses support the further evaluation of this EEHV vaccine in young, immunologically naïve elephants, which are the target population for protection against EEHV-HD through vaccination.

## Method
### Vaccine generation
**Production of recombinant (r)MVA-EE2-MCP.** A recombinant EEHV-MVA vector containing the genes for EE2 and MCP (NCBI accession numbers: YP_007969762.1, YP_007969788.1) was produced by the Viral Vector Core Facility (Jenner Institute/Pandemic Sciences Institute, University of Oxford, UK). The EE2 and MCP genes were codon optimised for vaccinia virus and the absence of TTTTNT pox virus termination motif. Genes were synthesised by Eurofins Genomics (Wolverhampton, UK) with 15 bp homology arms to the 5' and 3' end of the F11 or B8 insertion loci.

To generate rMVA viruses expressing EE2 under the control of the MVA wild-type (wt) B8 promoter and MCP under the control of the MVA wt F11 promoter[76], we employed a shuttle vector system and a stepwise transient dominant selection method of recombination[77]. The shuttle vectors contain a 1000 bp 5' flank consisting of the F11 or B8 promoter and a 3' flank consisting of 1000 bp of wt viral sequence 3' of the viral F11 or B8 protein coding sequence, as well as, outside of the insertion site flanking sequences, a green fluorescent protein (GFP) expression cassette for selection. The shuttle plasmids were amplified using F11 flinv F/R or B8 flinv F/R to create a linear DNA fragment which could be used for cloning. Primer sequences used for the generation of rMVA-EE2-MCP are listed in Supplementary Table 1. MCP and EE2 open reading frames (ORF) were digested from the Eurofins Genomics plasmids using KpnI and NotI unique restriction sites (New England Biolabs, Hitchin, UK). The ORF were inserted into the shuttle vectors using NEBuilder HiFi DNA Assembly mix (New England Biolabs) utilising the 15 bp homology arms to the 5' and 3' end of the F11 or B8 insertion loci.

Briefly, stepwise transient dominant selection to produce EEHV-MVA vector was performed as follows: The B8 shuttle vector[77] containing EE2 was recombined in DF-1 cells (American Type Culture Collection, Manassas, VA, USA) with a parental virus, MVA-F11 mCherry B8-BFP, which expresses two reporter genes, mCherry driven from the F11 promoter and blue fluorescent protein (BFP) driven from the B8 promoter. The cell lysate from this recombination was harvested and used to infect DF-1 cells. Cells were MoFlo single-cell sorted into 96-well plates based on the presence of GFP, mCherry and BFP. Virus plaque picking was performed until the virus culture was free of parental virus, and fully resolved as determined by the presence of mCherry reporter protein only. This intermediate parental virus was purified by sucrose cushioned centrifugation and the genotype checked by PCR (EE2 F/R and B8Fl F/R). The F11 shuttle vector[77] containing MCP was recombined with this intermediate parental virus as above selecting for the presence of GFP and mCherry. Virus plaque picking was performed until only colourless virus was present. Virus stocks were grown and purified using sucrose cushion centrifugation. The correct insertion of EE2 and MCP was verified by PCR (MCP F/R, EE2 F/R, F11Fl F/R, B8Fl F/R). Absence of parental MVA was determined by PCR (Purity F/R & B8Fl F/R). Virus was titrated by plaque assay.

**Production and purification of recombinant proteins.** Production and purification of recombinant antigens was carried out by Oxford Expression Technologies Ltd (OET; Oxford, UK). The full-length EEHV-1A EE2 amino acid (aa) sequence containing a C-terminal polyhistidine (6 × His)-tag was cloned into the transfer vector pOET1 (OET). For cloning of the MCP transfer vector, the first 215 aa of the N-terminal MCP sequence were truncated and replaced by a glutathione-S-transferase (GST)-tag. The transfer vectors were co-transfected with *flash*BAC™ ULTRA (FBU) viral DNA (OET) to generate recombinant baculoviruses FBU EEHV-EE2 and FBU EEHV- MCP$^{216-1349aa}$ in Sf9 insect cells (Fisher Scientific, Loughborough, UK). For recombinant protein production, Tni High Five cells (Fisher Scientific), 1 L at $1 \times 10^6$ cells ml$^{-1}$, were infected with FBU-EEHV-EE2 or FBU-EEHV-MCP$^{216-1349aa}$ virus stocks at 5 MOI. Cells were harvested 72 h post-infection and stored at −80 °C until processing.

To purify FBU-His-EEHV-EE2, cell pellets were incubated in lysis buffer (20 mM Tris-HCl, pH 7.8, 0.5 M NaCl, 10 mM imidazole, 6 M guanidine hydrochloride (GdmCl)) for 15 min on ice. The soluble fraction was clarified by centrifugation ($75,000 \times g$, 20 min, 4 °C) and incubated with 50% Ni-Sepharose suspension (GE Healthcare, Chalfont St Giles, UK) for 2 h at 4 °C with gentle rotation prior to affinity purification. Following incubation, the resin was washed thrice on a midi batch purification column (Generon, Maidenhead, UK) using wash buffer (20 mM Tris-HCl, pH 7.8, 0.5 M NaCl, 5 M urea, 10-50 mM imidazole) before elution in buffer containing 20 mM Tris-HCl, pH 7.8, 0.5 M NaCl, 300 mM imidazole and 5 M urea. Eluted FBU-His-EEHV-EE2

samples were analysed by SDS-PAGE. Suitable fractions were pooled, concentrated with Amicon Ultra 10 kDa filters (Merck Life Science Ltd, Gillingham, UK) and diluted in 20 mM Tris-HCl, pH 7.8, 0.5 M NaCl and 5 M urea buffer for storage at -80 °C.

For FBU-GST-EEHV-MCP[216-1349aa], the affinity purification process was similar, but the soluble lysate fraction was applied to a GSTrap FF 1 ml column (GE Healthcare). Elution steps were performed in an according buffer (50 mM Tris-HCl, 140 mM NaCl, 2 mM DDT, 5-10 mM reduced glutathione, pH 8.0). Total, applied and flow through fractions were prepared for further analysis and storage as above.

For our subunit vaccine formulations, we aimed to obtain a recombinant protein purity of ≥90%; however, this could not be achieved by affinity purification alone and the amount of rMCP protein recovered by this purification method was low. Therefore, affinity-purified rEE2 eluates and rMCP insoluble fractions were further subjected to size exclusion chromatography (SEC) purification: For rEE2, suitable eluates were further purified using Superdex 200 increase 10/300 or 16/600 columns (Cytiva, Little Chalfont, UK) on a GE AKTA Explorer system (GE Healthcare). The SEC was run using 20 mM Tris-HCl, pH 7.8, 0.5 M NaCl, 5 M urea buffer. To improve purification of rMCP, insoluble pellet fractions were resuspended in incubation buffer (20 mM Tris-HCl, pH 7.4, 0.5 M NaCl, 6 M GdmCl) on ice for 20 min. The soluble clarified fraction was recovered by centrifuging at $75,000 \times g$ for 20 min at 4 °C and applied to a Superdex 200 26/600 column (Cytiva) for SEC, which was run as above. Resulting eluted fractions were then subjected to another round of SEC as before.

rEE2 and rMCP fractions that yielded lower purities were used in whole blood stimulation assays and plasma antibody detection by western blot.

**Preparation of vaccine formulations.** rMVA-EE2-MCP vaccine formulations were prepared by diluting viral vector stocks in PBS with doses at $1 \times 10^8$ plaque-forming units (PFU) or $5 \times 10^8$ PFU.

For the preparation of 1 ml adjuvanted protein subunit vaccine, 25 µg or 50 µg of each rEE2 and rMCP antigen (50 µg or 100 µg total antigen) were mixed with Montanide Incomplete Seppic Adjuvant (ISA) 201 VG (SEPPIC, Paris, France) to produce a water-in-oil-in-water (W/O/W) vaccine emulsion. Briefly, pre-warmed ($35 \pm 1$ °C) adjuvant and aqueous antigen phases (55/45 v/v) were emulsified using a two-syringe system in accordance with the manufacturer's recommendations. Following emulsification, the W/O/W formulation was collected into one of the syringes and cooled at 4 °C for 1 hour before transfer to a 2-ml vial and storage at 2–8 °C until use.

## First-in-elephant, proof-of-concept vaccination study
### Study design
**Study elephants.** Equivalent to phase I vaccine trial standards, healthy adult elephants rather than EEHV-susceptible calves were chosen for this first-in-elephant proof-of-concept study. Accordingly, three adult animals ($N = 3$; elephant A: male, 21 years; elephant B: female, 18 years; elephant C: female, 49 years) were vaccinated and monitored for any adverse reactions. All three elephants had a history of EEHV detection and pre-existing humoral immunity, consistent with latent infection in adult animals prior to the study[42]. None of the elephants had any record of prior MVA-based vaccination in the context of cowpox prevention in European zoos.

The zoo housed the elephants under a "protected contact" system, where, in brief, interactions between animals and caretakers occur through secure barriers, contingent on the animal's voluntary participation during procedures such as routine care and health assessment, including blood sampling. Hence, animals in protected contact are trained to voluntarily participate in such procedures without the need for physical restraint. This system in general allows

for visual, auditory, and limited tactile contact, prioritising both animal welfare and handler safety.

**Vaccination and sampling regimen.** Pre- and post-vaccination blood samples were collected using a 23 G butterfly needle on a vacutainer collection system from either the caudal ear veins or the medial metatarsal vein under trained behavioural restraint. All vaccinations in this study were administered by hand to a depth of 1.5 inches (3.81 cm, needle hub) into the lower gluteal or proximal biceps femoris using a 22 G needle and 2-ml luer lock syringe under trained behavioural restraint.

Safety and immunogenicity testing of the EEHV prototype vaccines was carried out in a two-phased approach (Fig. 1):

In stage I ($n = 1$), elephant A received an initial prime-boost cycle of $1 \times 10^8$ PFU rMVA-EE2-MCP prime vaccine and 25 µg of rEE2 and rMCP each as adjuvanted subunit boost vaccine. The vaccinations were given at a 3-week interval. Elephant A was closely monitored for potential adverse reactions. Based on these observations, the first vaccination cycle was followed by a second prime-boost cycle 3 weeks later with $5 \times 10^8$ PFU of rMVA-EE2-MCP and 50 µg of each antigen in the rEE2-rMCP-subunit boost vaccine, also 3 weeks apart. Baseline blood samples for in vitro whole blood stimulation to measure immune responses to the vaccines were taken 2 weeks and 1 week prior to the first prime vaccination in stage I. Samples were also obtained before injection on each vaccination day and 1 week after each vaccination (Fig. 1a).

Two further elephants ($n = 2$; B and C) subsequently received the prime-boost vaccine at the higher doses. In addition, vaccination intervals in stage II were set to 4 weeks between prime and boost vaccinations and 6 weeks between the first and second vaccination cycle. Baseline blood samples were collected in the weeks prior to the first vaccination. Blood was further sampled before immunisation on vaccination days, 2 and 3 weeks after each prime vaccination and 2, 3 and 4 weeks post-boost vaccinations (Fig. 1b). Following each blood sampling, lithium heparin-anticoagulated whole blood aliquots were prepared for immediate in vitro whole blood stimulation.

### Elephant observation and vaccine safety assessment
Routine observations of the study animals were made daily. Observations of the elephants' physical and behavioural conditions were made during one morning and one afternoon direct protected contact interaction period with each elephant. During these sessions, keepers conducted close observations to assess the animals' condition, examining their physical state as they moved around. This included inspecting each individual's mouth, teeth, eyes, tail, and overall body for any health observations.

Post-vaccination monitoring occurred after each injection. There was an initial observation period of at least 30 minutes where both the elephant keeper and veterinarian continuously monitored the study elephant for any immediate reactions, either at the injection site or in behaviour. Once satisfied, the individual was reintroduced to the herd, and further visual checks were carried out from a distance by keepers throughout the day. Most administrations occurred during the morning husbandry session, allowing for additional close examination during the day.

Any changes in physical or behavioural state were recorded in the daily reporting documentation and relayed to the veterinary department if not already present. Periodic communications between the elephant team and veterinary team occurred in the days following an administration, with a veterinarian also conducting live observations while the elephants were at a distance within the enclosure.

### Immunogenicity assessment
**Vaccine antigen-specific whole blood stimulation assays.** We used a modified IFNG release (IGRA) assay protocol for on-site, antigen-

specific stimulation of small whole blood volumes followed by RT-qPCR in the laboratory. This approach was based on existing protocols for HCMV-specific IGRA with ELISA read-out[78,79], HCMV-specific ELISpot assays[80] and on protocols for small-volume whole blood stimulation assays developed for determining SARS-CoV-2-specific T cell responses by qPCR[81,82]. For this, 300 µl of lithium heparinised blood was transferred to 2-ml tubes containing rEE2 or rMCP antigens, which had been prediluted in 50 µl PBS to give a final assay concentration of 10 µg ml⁻¹. Tubes containing PBS alone served as unstimulated controls, and pokeweed mitogen was used as a positive control to verify cell viability and responsiveness. The assays were mixed by pipetting and incubated at 37 °C for 20 h. Whole blood stimulation was terminated by adding 1.3 ml of RNA*later* Stabilization Solution (Thermo Fisher Scientific, Altrincham, UK). RNA-preserved blood samples were sent to the laboratory at room temperature where they were stored at -20 °C until downstream processing.

**RNA extraction and *IFNG* RT-qPCR.** Whole blood samples in *RNAlater* were centrifuged at $5000 \times g$ for 5 min followed by aspiration of the stabilisation solution. The remaining cell pellets were homogenised in 1050 µl; TRIzol Reagent (Thermo Fisher Scientific) according to the manufacturer's instructions, and 210 µl chloroform (Sigma Aldrich, Gillingham, UK) was added to each sample to obtain an aqueous phase. RNA was subsequently extracted from the aqueous phase using the RNeasy MinElute Cleanup Kit (Qiagen, Manchester, UK) based on the manufacturer's recommendations but with an additional 80% ethanol wash step prior to column membrane drying. RNA concentration and quality were measured using a Nanodrop 2000 spectrophotometer (Agilent Technologies, Stockport, UK). Eluates were stored at -80 °C until further processing for RT-qPCR.

Invitrogen SuperScript IV VILO Master Mix (Thermo Fisher Scientific) was employed to convert 250 ng of total RNA into cDNA as per the manufacturer's instructions, including prior genomic DNA removal with kit-supplied ezDNase.

Elongation factor-1α (*EF1A*) served as a reference control in RT-qPCR. Primers for *EF1A* and *IFNG* were taken from refs. 83,84, respectively, and sequences were as follows: *EF1A* forward primer: 5′-AGTCTGTTGAAATGCACCCC-3′; *EF1A* reverse primer: 5′-GCTA CATTGCCACGACGAAC-3′; *IFNG* forward primer: 5′-GGAA TATCTTAATGCAACTGATTCA-3′; and *IFNG* reverse primer: 5′-CCTGGTTGTCTTTCAAGTTGTCAA-3′. RT-qPCR reactions were performed with Brilliant III Ultra-Fast SYBR Green QPCR Master Mix on an AriaMx Real-time PCR System (both Agilent Technologies). For relative expression analysis, the comparative $2^{-(\Delta\Delta Ct)}$ method was applied to normalise *IFNG* expression to the endogenous *EF1A* reference and to express modulation as the relative FC to PBS controls.

**Transcriptomic analysis.** RNA samples from the stage II trial were processed for RNA-seq with the aim to elucidate differential gene expression before and after vaccination in unstimulated and antigen-stimulated transcriptomes beyond the expression of *IFNG*. Pre-vaccination samples from time points W-1 and W0 were compared with samples at W14 and W17 (elephant C)/W18 (elephant B), following vaccination rounds.

For sample processing, equal amounts of rEE2- and rMCP-stimulated RNA samples of the respective time points were pooled. RNA quality was assessed by RNA/DNA quantification with a Qubit fluorometer (Thermo Fisher Scientific) and a Fragment Analyzer (Agilent Technologies). All samples were found to have good quality (RQN >8). RNA-seq libraries of the total RNA eluates from unstimulated and antigen-stimulated blood were prepared at Genewiz-Azenta (Leipzig, Germany). The NEBNext Ultra Directional RNA Library Prep Kit for Illumina was utilised in conjunction with the NEBNext rRNA Depletion Kit (New England Biolabs) to perform ribo- and globin-depletion-based library preparation. Sequencing was carried out on a NovaSeq 6000 instrument (Illumina, San Diego, CA. USA) to produce paired-end reads.

Reads obtained from the blood samples described above were submitted to differential gene expression analysis across three comparisons: A) unstimulated blood samples before vaccination vs. unstimulated samples after vaccination; B) blood samples stimulated with vaccine antigens before vaccination vs. unstimulated samples before vaccination; and C) blood samples stimulated with vaccine antigens after vaccination vs. unstimulated samples after vaccination (Supplementary Fig. 1). For comparison A, an unpaired moderated Student's t-test was employed, while paired moderated Student's t-tests were used for comparisons B and C. These moderated t-tests in DESeq2 are shrinkage-based two-sided Wald tests ($H_0$: $\log_2 FC = 0$), which incorporate empirical Bayes moderation to improve dispersion estimation across genes, allowing for more reliable inferences[85]. Sequencing adapters were trimmed following quality checks on the reads performed using FastQC[86] (v0.11.9). The reads were subsequently aligned to the *Elephas maximus indicus* reference genome (annotation release GCF_024166365.1-RS_2023_02) using the Subread package[87] and quantified using featureCounts[88] (both v2.0.6). Quality control, normalisation and statistical analyses were conducted using the DESeq2 package (v1.44) in R/Bioconductor[85] (v4.3.2 and v3.19, respectively), with paired comparisons applied where relevant. Adjusted $p$ values were calculated using the Benjamini-Hochberg method for multiple testing control. DEG were identified based on the following thresholds: padj ≤0.05 for comparison A and padj ≤0.01 for comparisons B and C, with absolute fold-change values ≥1.25. Venn Diagrams to obtain lists of gene expression changes and enriched pathways specific to each group were created with the online tool jvenn (https://jvenn.toulouse.inrae.fr/app/index.html)[89].

Functional ORA were conducted using the Enrichr analysis tool[90]. The statistical significance was calculated using a one-sided Fisher's exact test, evaluating the over-representation of functional gene sets against the null hypothesis of no association, with multiple comparison correction applied using the Benjamini-Hochberg method. Resulting terms were ranked by the combined score. For comparison A, the integrated Human Gene Atlas database[91] (release not versioned; accessed 2024) was used to identify cell type enrichment terms, and pathways with a $p$ value ≤0.05 were considered statistically significant. For comparisons B and C, ORA was performed using the GOBP database[92] (release 2023) and the Reactome Pathway Knowledgebase[93] (release 2022) to identify over-represented pathways in the upregulated gene sets (padj ≤0.05 and padj ≤0.01, respectively). To reduce redundancy in GOBP ORA, GO terms with more than 85% gene overlap were eliminated, ensuring a Jaccard index close to 1.

**Plasma antibody detection by western blot.** Although the vaccine was primarily designed to elicit T cell responses, western blot was carried out for the detection of antibodies against rEE2 and rMCP antigens using pre- and post-vaccination plasma samples. Recombinant proteins were separated using Bolt 4–12% Bis-Tris or NuPAGE 3–8% Tris-Acetate gels and transferred to nitrocellulose membranes with the iBlot 2 Dry Blotting System (all Thermo Fisher Scientific). Membranes were blocked and probed with a 1:50 dilution of elephant plasma using the iBind Flex Western System (Thermo Fisher Scientific). HRP-conjugated Protein A/G (1:5000) and SuperSignal West Pico PLUS substrate (Thermo Fisher Scientific) were used for detection and signals visualised using an Azure c280 imaging system (Cambridge Bioscience, Cambridge, UK). The resulting data are shown in Supplementary Fig. 2.

**Statistics & reproducibility.** This study was designed as a proof-of-concept study and conducted as a single trial on a small cohort of animals from a protected species. The sample size was therefore

determined by the number of Asian elephants that were available and suitable for inclusion in a study focussed on vaccine safety. Due to the endangered status of the species, protected contact husbandry and welfare considerations, biological replication in this single trial was limited. Accordingly, no statistical method was used to predetermine sample size. No data were excluded from the analyses, except for three samples that did not meet downstream quality checks, as indicated in Fig. 2. The experiments were not randomised. Blinding was not possible in line with 3 R animal review ethics.

## Ethical statement

Ethical approval of this study was granted by the Home Office of the United Kingdom (PP1309593) and the Animal Welfare and Ethical Review Body (AWERB) at the Animal and Plant Health Agency (APHA) and by the North of England Zoological Society (Chester Zoo) ethical review committee. All animal procedures were carried out in accordance with the United Kingdom Animals (Scientific Procedures) Act of 1986.

## Reporting summary

Further information on research design is available in the Nature Portfolio Reporting Summary linked to this article.

## Data availability

The RNA-seq data reported in this study were deposited in the NCBI Gene Expression Omnibus Database under accession number GSE297105. All other data related to the findings of this study are available in the Article and its Supplementary Data and Supplementary Information. Source data for RT-qPCR gene expression analysis and unprocessed western blot images are provided in the Source Data file. Source data are provided with this paper.

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

## Acknowledgements

The authors would like to thank Claire Powers at the Viral Vector Core Facility, University of Oxford, for assistance in generating and supplying the rMVA-EE2-MCP construct. We thank Hugh Simmons and the Animal Sciences Home Office Compliance team at APHA for assistance with managing the Animals (Scientific Procedures) Act aspects of the animal study. We are grateful to Andra Corder and Sebastien Deville of SEPPIC, Air Liquide Healthcare, for the provision of Montanide ISA 201 VG and their expert technical guidance. T.M. was supported through Chester Zoo's Conservation and Science Scholars and Fellows Programme. Studies were supported by grant RC7518 from Chester Zoo and a University Global Partnership Network-UGPN grant at the University of Surrey to F.S. H.I.N. received funding from the Fundação de Amparo à Pesquisa do Estado de São Paulo-FAPESP (grant No. 2018/14933-2 and 2019/16444-1) and the Brazilian National Council for Scientific and Technological Development (grant No. 1).

## Author contributions

T.M. Conceptualisation, data curation, formal analysis, investigation, methodology, project administration, resources, supervision, visualisation, writing—original draft, writing—review & editing. J.L. Conceptualisation, investigation, methodology, project administration, resources, supervision, writing—review & editing. G.D. Investigation, methodology, project administration, resources, writing—review & editing. F.M.F. Data curation, formal analysis, resources, visualisation, writing—review & editing. R.F. Investigation, methodology, project administration, resources. R.M. Investigation, resources. R.K. Methodology, resources, writing–review & editing. S.M. Methodology, resources, writing—review & editing. A.C. Methodology, resources, writing—review & editing. L.P.G. Methodology, resources, writing–review & editing. S.L.W. Conceptualisation, funding acquisition, supervision. A.D. Conceptualisation, formal analysis, writing–review & editing. K.L.E. Funding acquisition, supervision, writing—review & editing. H.I.N. Formal analysis, resources, visualisation, writing–review and editing. F.S. Conceptualisation, formal analysis, funding acquisition, methodology, project administration, resources, supervision, visualisation, writing—review & editing. Roles defined by the Contributor Roles Taxonomy, available at (https://credit.niso.org/).

## Competing interests

The authors declare no competing interests.
