## [Transparent Peer Review file · Nature Communications]

A safe, T cell-inducing heterologous vaccine against elephant endotheliotropic herpesvirus in a first-in-elephant proof-of-concept study

Corresponding Author: Professor Falko Steinbach

Version 0:

Reviewer comments:

Reviewer #1

(Remarks to the Author)

Elephant endotheliotropic herpesvirus haemorrhagic disease (EEHV-HD) is a significant cause of calf and juvenile mortality in semi-domesticated and free-ranging Asian elephants and African elephants. The absence of a viral culture and a limited toolbox for the elephant immune system relative to other animal species has hampered progress in antiviral drug and vaccine development. The development of an EEHV-1A vaccine is a significant contribution to save this endangered species. The findings of this first-in-elephant proof-of-concept EEHV vaccine trial demonstrate that heterologous prime boost vaccines can be safely used in this species. The methods used are in details following all necessary steps and adequate with the objectives of the study. The tools to study immune response against vaccine in elephant are not available. Development of non-replicating safe virus vector and protein cocktail boost component showed meticulous design in development of the immunogenic candidate. Analysis of immune responses by RT-qPCR gene expression in pre and post-immunization study is a novel approach in absence of non-available cytokine assay tool in elephant. Again development of point of care IFNG assay technique is an additional outcome of the study to evaluate CMI response in the study site. The study meticulously included top ranked immune genes as well as a substantial number of other immune-relevant genes in both pre- and post-vaccination study.

The claims of the study is well supported by research findings. However, there is sufficient critical analysis on T cell mediated immune response with involvement of cytotoxic T cells. The study could be more data based with involvement of more number of elephants of different age groups in vaccination study with and without EEHV latent infection. In future tools should be optimized for assay of cytokines and typing of lymphocyte subsets.

Reviewer #2

(Remarks to the Author)

A vaccine against EEHV is a major need and would be a great achievement. Here Maehr et al investigates safety and some immunogenicity of a prime-boost vaccination of a rMVA vaccine vector containing major capsid protein (MCP) and putative regulatory protein EE2 followed by the same proteins formulated in a w/o/w adjuvant as subunit vaccine.

The vaccine development is state-of-the-art and described sufficiently. The vaccine is described for EEHV1A so a comment on how conserved the antigens are between EEHV1A and EEHV1B/EEHV4 would be appropriate?

The primary goal is vaccine safety which is evaluated by vaccinating three adult elephants in a good study design with low dose vaccinations in a single elephant followed by standard dose vaccination of the same elephant plus two additional elephants. Given safety is the primary aim, the reporting is very scarce. Even though no safety concerns were observed, the authors must present more on how the observations were made and what they specifically were looking for. The current description "The elephants were visually assessed throughout the day, every day for the duration of the trial, monitored for any changes in behavior or physical changes at the sites of infection, and findings recorded" is too short and non-descriptive. This could mean anything from i) observations were randomly noted by zoo personnel during ordinary work if they observed anything to ii) a protocol with a dedicated person following a check list with pre-specified grading system on e.g. behavior, social interaction, appetite, temperature and palpation of injection site for observation of pain with a certain frequency decided from the time since vaccination. Also, were the blood samples used for any read-outs on blood

parameters to support the good safety reported?

Much more detail is given to the secondary goal of immunogenicity studies initially concentrating on antigen-specific IFN-gamma response following a nice bedside stimulation assay with RT-qPCR as read-out. Were there any positive control stimulations to validate the efficacy of bed-side stimulation? The study benefits from several pre-immunization samples to obtain more confidence in the pre- to post-vaccination read-outs when the elephants appear to be latently infected with EEHV. The latent infection is however only circumstantial based on what is expected and the IFN γ response to the vaccine antigens. The study would benefit from additional read-outs on EEHV serology if that is possible and certainly for antibody responses to vaccine antigens and the rMVA vector.

Immunogenicity in naïve proxy animals (pigs?) would validate the vaccines immunogenicity. It is hard to believe this has not been performed prior to elephant studies...

A lot of interesting and detailed analyses are performed on the scRNAseq with appropriate ORA to group responses into GO terms. This is a new level of data for elephant immunology, but in terms of the relation to the vaccine a discussion on the rMVA alone should be included. Currently it is presented as EEHV-vaccine responses, but although cells were stimulated with the proteins some, or most, of the observations are likely more related to the live MVA vector than the selected antigens. Elephant studies are obviously not easy to perform and should be well justified, but studies (alternatively in a proxy species) with empty MVA vector, rMVA without w/o/w (and vice versa) would add another dimension to the understanding of the vaccine immunogenicity. This should at least be discussed.

Minor comments:

ORA is not explained as an abbreviation (on p 11)

Page 14 second last paragraph: "revealed" should be replaced by "indicate"

Page 16 something is missing in front of "range countries"

Page 17 first paragraph. Include a sentence of possible replication and shedding of the MVA vector in relation to safety profile.

Page 23 last paragraph. With more than one sample for each elephant prior to vaccination, how is a t-test the right analysis?
Gregers Jungersen

Reviewer #3

(Remarks to the Author)

The authors present the novel findings and the first report of EEHV vaccine trial in adult Asian elephants. The sequence of the manuscript is easy to follow and well written with showing an alternative immunological diagnostic tool for EEHV vaccine research. Currently, there are several research groups have being investigated the EEHV vaccine trails in either adult or juvenile Asian elephants with no report yet. Despite the manuscript being interesting and suited to publication, there are some points that need to be addressed or discussed prior to publication.

Major point:

1. The major limitation of the present study is the number of animal samples used. Despite it being understood in case of experiments in such large, endangered animals like elephants, the previous history of animals showing the previous history of EEHV(s) exposure, immune status against EEHV major proteins, such as glycoproteins, should be addressed and mentioned in the Materials and Methods and result section.

2. Although the authors aimed to investigate the cellular immune response using the EE2 and MCP internal proteins, brief comparison and discussion of this vaccine platform with other previous surface envelope glycoprotein platforms should also be mentioned in the discussion section.

Minor point:

1. Why the authors choose to investigate the heterologous vaccine platform for the first time in adult elephants instead of homologous vaccines. This hypothesis and theory should be mentioned in the introduction section.

Version 1:

Reviewer comments:

Reviewer #2

(Remarks to the Author)

The revised manuscript has addressed my concerns and it is now suitable for publication.

Reviewer #3

(Remarks to the Author)

Reviewer #1 (Remarks to the Author)

Elephant endotheliotropic herpesvirus haemorrhagic disease (EEHV-HD) is a significant cause of calf and juvenile mortality in semi-domesticated and free-ranging Asian elephants and African elephants. The absence of a viral culture and a limited toolbox for the elephant immune system relative to other animal species has hampered progress in antiviral drug and vaccine development. The development of an EEHV-1A vaccine is a significant contribution to save this endangered species. The findings of this first-in-elephant proof-of-concept EEHV vaccine trial demonstrate that heterologous prime boost vaccines can be safely used in this species. The methods used are in details following all necessary steps and adequate with the objectives of the study. The tools to study immune response against vaccine in elephant are not available. Development of non-replicating safe virus vector and protein cocktail boost component showed meticulous design in development of the immunogenic candidate. Analysis of immune responses by RT-qPCR gene expression in pre and post-immunization study is a novel approach in absence of non-available cytokine assay tool in elephant. Again development of point of care IFNG assay technique is an additional outcome of the study to evaluate CMI response in the study site. The study meticulously included top ranked immune genes as well as a substantial number of other immune-relevant genes in both pre- and post-vaccination study.

The claims of the study is well supported by research findings. However, there is sufficient critical analysis on T cell mediated immune response with involvement of cytotoxic T cells. The study could be more data based with involvement of more number of elephants of different age groups in vaccination study with and without EEHV latent infection. In future tools should be optimized for assay of cytokines and typing of lymphocyte subsets.

Authors' Comments

We appreciate Reviewer #1's thoughtful review and the positive remarks regarding our vaccine's study. We hope to be able to follow this up in the way the reviewer proposes, i.e. with more elephants and more analysis of cellular subsets, also expanding the elephant immunobiology knowledge and toolbox further. We trust that our initial proof-of-concept study has paved the way for such future studies.

Reviewer #2 (Remarks to the Author)

A vaccine against EEHV is a major need and would be a great achievement. Here Maehr et al investigates safety and some immunogenicity of a prime-boost vaccination of a rMVA vaccine vector containing major capsid protein (MCP) and putative regulatory protein EE2 followed by the same proteins formulated in a w/o/w adjuvant as subunit vaccine.

Authors' Comments

We thank the reviewer for the thorough assessment and for raising valuable considerations that will improve the manuscript. We reply below point by point and propose changes to the manuscript accordingly.

The vaccine development is state-of-the-art and described sufficiently. The vaccine is described for EEHV1A so a comment on how conserved the antigens are between EEHV1A and EEHV1B/EEHV4 would be appropriate?

Authors' Comments

We carried out a phylogenetic comparison of the two proteins (EE2 and MCP) using all viruses described to cause disease in Asian elephants, which include next to EEHV-1A and EEHV-1B, EEHV-4 and EEHV-5. The homology tables provided in Supplementary Fig. 3 support the view that for both proteins the sequence identity overall and in large parts of the proteins would suffice to elicit a cross-reactive response for EEHV-1A and -1B. In case of MCP, but not EE2 the homology in parts might also suffice to lead to some, but limited, cross-reactivity with EEHV-5. EEHV-4 is the most distant of these EEH viruses and we would not assume any cross-protection from the vaccine for this genotype. We have accordingly added a respective comment into the discussion of the manuscript (pp. 17–18, lines 487–494).

The primary goal is vaccine safety which is evaluated by vaccinating three adult elephants in a good study design with low dose vaccinations in a single elephant followed by standard dose vaccination of the same elephant plus two additional elephants. Given safety is the primary aim, the reporting is very scarce. Even though no safety concerns were observed, the authors must present more on how the observations were made and what they specifically were looking for. The current description “The elephants were visually assessed throughout the day, every day for the duration of the trial, monitored for any changes in behavior or physical changes at the sites of infection, and findings recorded” is too short and non-descriptive. This could mean anything from i) observations were randomly noted by zoo personnel during ordinary work if they observed anything to ii) a protocol with a dedicated person following a check list with pre-specified grading system on e.g. behavior, social interaction, appetite, temperature and palpation of injection site for observation of pain with a certain frequency decided from the time since vaccination. Also, were the blood samples used for any read-outs on blood parameters to support the good safety reported?

Authors' Comments

We appreciate that this part of our description was not sufficiently clear and we modified the respective section in the Methods to provide a clearer and more comprehensive account of the observations carried out. In this context, we would like to highlight that modern zoological institutions manage elephants under a ‘protected contact’ system, where interactions between animals and caretakers occur through secure barriers. This system allows for visual, auditory, and limited

tactile contact during procedures such as health assessments, contingent on the animal's voluntary participation. Except in specified emergencies, personnel managing elephants are no longer allowed to enter the enclosures to directly interact with or restrain the animals. This approach prioritises both animal welfare and handler safety. Accordingly, while animals are trained to provide blood (or other) samples, this is not without obstacles and sampling frequency and size can be limited. In the manuscript's Methods section, we have added a brief section explaining protected contact husbandry on p. 23, lines 661–667 and introduced a new “*Elephant observation and vaccine safety assessment*” section on pp. 24–25, lines 694–712, which outlines the observation protocol as follows:

"Routine observations of the study animals were made daily. Observations of the elephants' physical and behavioural conditions were made during one morning and one afternoon direct protected contact interaction period with each elephant. During these sessions, keepers conducted close observations to assess the animals' condition, examining their physical state as they moved around. This included inspecting each individual's mouth, teeth, eyes, tail, and overall body for any health observations.

Post-vaccination monitoring occurred after each injection. There was an initial observation period of at least 30 minutes where both the elephant keeper and veterinarian continuously monitored the study elephant for any immediate reactions, either at the injection site or in behaviour. Once satisfied, the individual was reintroduced to the herd, and further visual checks were carried out from a distance by keepers throughout the day. Most administrations occurred during the morning husbandry session, allowing for additional close examination during the day.

Any changes in physical or behavioural state were recorded in the daily reporting documentation and relayed to the veterinary department if not already present. Periodic communications between the elephant team and veterinary team occurred in the days following an administration, with a veterinarian also conducting live observations while the elephants were at a distance within the enclosure.”

No other blood parameters such as cortisol, inflammatory cytokines, or acute phase proteins were measured. Those were deemed insufficient to assess vaccine safety due to the lack of published literature establishing what constitutes a 'normal' versus altered biological response.

We believe that this additional information on our observation protocol under protected contact strengthen the case for vaccine safety.

Much more detail is given to the secondary goal of immunogenicity studies initially concentrating on antigen-specific IFN-gamma response following a nice bedside stimulation assay with RT-qPCR as read-out. Were there any positive control stimulations to validate the efficacy of bed-side stimulation?

Authors' Comments

We thank the reviewer for the appreciation of the experimental setup that we believe could be widely used for similar purposes in wildlife species well beyond the Asian elephants described here.

Indeed, we did evaluate the reactivity (viability) of the blood samples through pokeweed mitogen (PWM), which is known to elicit T and B cell activation and used *IFNG* as a readout.

Below we present one example from elephant C. For clarity and readability of the manuscript graphs, we believe that inclusion of PWM positive control data in the main manuscript is not beneficial as it is a technical point only. In field applications this would be important to ensure the samples were

actually viable when tested. Accordingly, we now document the use of PWM as a positive control for cell viability and responsiveness in the Materials and Methods section (p. 25, lines 723–724).

Elephant C

Data from Fig. 2e and 2f with the inclusion of pokeweed (PWM) positive control stimulation. The relative expression of *IFNG* mRNA, normalised to the reference gene elongation factor 1 α , is shown. Controls (PBS) were set to 1 based on the $2^{(-\Delta\Delta Ct)}$ calculation, and all other samples are expressed relative to these controls. The longitudinal *IFNG* responses to PWM demonstrate the responsiveness of the elephant immune cells to stimulation.

The study benefits from several pre-immunization samples to obtain more confidence in the pre- to post-vaccination read-outs when the elephants appear to be latently infected with EEHV. The latent infection is however only circumstantial based on what is expected and the IFNg response to the vaccine antigens. The study would benefit from additional read-outs on EEHV serology if that is possible and certainly for antibody responses to vaccine antigens and the rMVA vector.

Authors' Comments:

We thank the reviewer for the appreciation of the experimental design and would like to address the two points made within this comment separately:

1) **Regarding latent EEHV infection**, we understand that the reviewer is aware of the textbook established latency of herpesviruses in the family *Orthoherpesviridae*, which includes HCMV, HHV-6 and -7 as nearest neighbours of EEHV. Additionally, there is ample evidence for EEHV latency (for example in references¹⁻⁵ listed below this response). Hence, we focus on addressing the specific status of the three animals in the study concerning their previous infection and latency here.

Following the reviewer's feedback, we have now integrated further information in both the Materials and Methods and the Results section to clarify that all three study animals were known to have had prior EEHV exposure and pre-existing humoral immunity consistent with latent infection. The relevant historical data supporting this was obtained by Chester Zoo independently of this study in the context of routine herd health management and as part of a BSc Honours dissertation at the University of Liverpool¹⁶ (42 in manuscript). Hence, the Materials and Methods section describing the study cohort on p. 23, lines 656–658, now includes the following:

"All three elephants had a history of EEHV detection and pre-existing humoral immunity, consistent with latent infection in adult animals prior to the study⁴²."

In the Results section, we revised the paragraph discussing baseline *IFNG* responses to highlight that the observed recall responses are supported by historical shedding and serological data (p. 6. lines 145–148). The sentence now reads:

“These recall responses before vaccination are supported by the notion that all three adult elephants were latently infected with EEHV as anticipated for adult animals, which is corroborated by historical EEHV shedding and serological data indicating humoral immunity to major viral glycoproteins⁴².”

Additionally, in the paragraph discussing the animals’ pre-existing immunity and exposure history on p. 14, lines 385–388 (Discussion section), we have added the above dissertation as an additional reference.

We trust that these additions substantiate that the evidence of latent infection obtained from our pre-vaccination *IFNG* read-outs is not “*only circumstantial based on what is expected and the IFNg response to the vaccine antigens*” but supported by Chester Zoo’s pre-existing health management records on the immune status and exposure history of the animals used in this study.

2.) Regarding antibodies to vaccine antigens, we provide serology results to the recombinant EEHV antigens to strengthen the manuscript as requested. As our vaccine was specifically designed to induce T cell-mediated responses and the antigens would not be suitable to induce neutralising antibody responses, we performed western blot analysis as an ancillary evaluation that may offer diagnostic value in the future. We agree with the reviewer that antibody detection holds merit for evaluating vaccine-induced immune activation, and for further optimising vaccination protocols.

Details of western blot antibody detection can be found in the Materials and Methods section (pp. 27–28, lines 791–800) and Supplementary Fig. 2 where we provide the longitudinal antibody monitoring of elephants B and C for which we also carried out transcriptomics analyses. We would not propose to include this into the main part of the manuscript since the biological value of these antibodies is limited and they are not by themselves protective.

In the Discussion, we embed this analysis in the paragraph discussing over-represented B cell-related pathways on p. 17, lines 469–476. This now reads as follows:

“Despite our vaccine being primarily designed to induce a T cell response, western blot analysis demonstrated that the vaccine elicited measurable humoral responses, with antibody levels increasing post-vaccination compared to baseline. A degree of humoral activation against the vaccine antigens was expected and the over-represented pathways fit well with the activation of Th cells providing help to B cells, as well as some pathways pointing directly towards a secondary activation of B cells. Although antibodies induced by our vaccine would be of limited use in the immune defence, ancillary serological data may in the future serve as a diagnostic marker of vaccination.”

Additionally, where discussing the animals’ pre-existing immunity and exposure on p. 14, lines 384–390, we have edited the paragraph to add corroborating detection of pre-vaccination antibodies to the vaccine antigens. This paragraph, with the addition in bold, now reads:

*“The analysis of cellular immune responses focussed initially on *IFNG* gene expression as a well-established marker for such. That all three elephants had pre-existing immunity due to their latent infections was not necessarily surprising since all adult elephants may be considered latent EEHV carriers⁵⁴ and our study cohort was previously exposed to multiple EEHVs, including the target EEHV-1A^{12, 42}. **Notably, low levels of antibodies against both EE2 and MCP were detectable in all elephants prior to vaccination (Supplementary Fig. 2), which corroborates latent infection.**”*

Antibody responses against the MVA vector were not determined. We had not considered an anti-vector response within the scope of this proof-of-concept study and believe that the reviewer's request may stem from a misunderstanding. In the manuscript's Introduction and Discussion (p. 5, lines 118–119 and p. 14, lines 376–377), we mentioned that MVA has previously been used in elephants as a vaccine against cowpox^{7, 8}. Therefore, MVA was already a tested vector in the target species that did not warrant additional safety testing in further animals in line with the 3R concept of animal experimentation.

However, we had not intended to imply that the elephants at Chester Zoo had been vaccinated using MVA before. In fact, none of the study elephants have any record of being vaccinated with MVA-based vaccine and we have added this information to the description of the elephants on p. 23, lines 658–660. Hence, we hope to have put across the argument why testing for the MVA vector response has not been warranted and that a pre-existing anti-MVA immunity could be excluded due to the lack of previous exposure. In our study, such an effect could only to a limited extent appear after the second MVA vaccination but would remain irrelevant for the EEHV-specific response. In this context, anti-vector immunity was also not of immediate concern since heterologous prime boost designs are known to mitigate such an effect^{9, 10}. Our anti-vaccine antigen read-outs demonstrate the generation of antigen-specific reactions since we only used recombinant proteins to determine both T and B cell responses. However, we acknowledge that a more detailed characterisation of responses to different vaccine components will be important in future studies.

Separately, we appreciate that the MVA vector may have immune-stimulatory capacity as we discuss in response to a subsequent point by the reviewer.

References cited in this response (excl. citations in manuscript excerpts):

1. Hardman, K. et al. Detection of elephant endotheliotropic herpesvirus type 1 in asymptomatic elephants using TaqMan real-time PCR. *Vet. Rec.* **170**, 205 (2012).
2. Ackermann, M., Hatt, J. M., Schetle, N. & Steinmetz, H. Identification of shedders of elephant endotheliotropic herpesviruses among Asian elephants (*Elephas maximus*) in Switzerland. *PLoS One* **12**, e0176891 (2017).
3. Kochagul, V. et al. Production of antibody against elephant endotheliotropic herpesvirus (EEHV) unveils tissue tropisms and routes of viral transmission in EEHV-infected Asian elephants. *Sci. Rep.* **8**, 4675 (2018).
4. Sree Lakshmi, P. et al. Pathological and molecular studies on elephant endotheliotropic herpesvirus haemorrhagic disease among captive and free-range Asian elephants in India. *Microb. Pathog.* **175**, 105972 (2023).
5. Srivorakul, S. et al. Possible roles of monocytes/macrophages in response to elephant endotheliotropic herpesvirus (EEHV) infections in Asian elephants (*Elephas maximus*). *PLoS One* **14**, e0222158 (2019).
6. Ward, E. *An investigation of elephant endotheliotropic herpesvirus (EEHV) shedding and viraemia in a herd of 10 Asian elephants (Elephas maximus) at a UK zoo in relation to births and deaths*. Intercalated Honours BSc dissertation in Veterinary Conservation Medicine, University of Liverpool (2023).
7. Pilaski, J. & Zhou, X. Die Pockenimpfung der Elefanten. *Verhandlungsbericht über die Erkrankungen der Zootiere* **33**, 203–211 (1991).
8. Mahnel, H. & Mayr, A. Erfahrungen mit der Immunisierung gegen Orthopockenviren von Menschen und Tieren unter Verwendung des Impfstoffstammes MVA. *Berl. Münch. Tierärztl. Wochenschr.* **107**, 253–256 (1994).
9. McCann, N., O'Connor, D., Lambe, T. & Pollard, A. J. Viral vector vaccines. *Curr. Opin. Immunol.* **77**, 102210 (2022).
10. Kardani, K., Bolhassani, A. & Shahbazi, S. Prime-boost vaccine strategy against viral infections: Mechanisms and benefits. *Vaccine* **34**, 413–423 (2016).

Immunogenicity in naïve proxy animals (pigs?) would validate the vaccines immunogenicity. It is hard to believe this has not been performed prior to elephant studies...

Authors' Comment

We fully appreciate the importance of utilising model systems in early-stage vaccine development. However, we may possibly disagree here due to the nature of the study and different 3R cultures being applied. In our environment (that is the local Animal Welfare and Ethical Review Body and the licensing authority, i.e. the Home Office in the UK) it would have been difficult to justify a safety study of a MVA vector vaccine or a protein subunit vaccine in any other species, given that such had been executed in many species and the proteins in question have no regulatory function that would have warranted such neither. As for immunogenicity, it is reasonable to predict that the vaccine components would elicit an immune response, at least to some extent, in other species based on numerous studies demonstrating this in principle.

However, when it comes to assessing immunogenicity in the context of EEHV, Asian elephants present a unique case in immunology that makes them exceptionally difficult to model. Their closest extant relatives are members of the orders Hyracoidea (hyraxes) and Sirenia (manatees and dugongs), wildlife species that are unsuitable as model organisms. More importantly, elephant blood features an extraordinary composition of immune cells in that their immune profile is defined by an unusually large monocyte population and a comparatively small lymphocyte compartment. Additionally, elephants, and Afrotheria more broadly, possess heterophils in place of neutrophils, a characteristic that further sets their immune system apart^{1,2}. Accordingly, results from standard models like pigs would not have added value in this context and given the current stage of knowledge about these platforms and their performance in elephants. We trust that these considerations sufficiently support our rationale for proceeding with in-elephant studies directly. We have added a statement in the discussion section that briefly raises awareness to the matter (p. 18, lines 495–502).

References cited in this response:

1. Weisbrod, T. C., Isaza, R., Cray, C., Adler, L. & Stacy, N. I. The importance of manual white blood cell differential counts and platelet estimates in elephant hematology: blood film review is essential. *Vet. Q.* **41**, 30–35 (2021).
2. Harr, K. E., Isaza, R. & Blue, J. T. Hematology of elephants. In *Schalm's Veterinary Hematology*, 6th edn. (eds. Douglass, J., Weiss, K. & Wardrop, J.) 942–949 (Wiley-Blackwell, Ames, Iowa, 2010).

A lot of interesting and detailed analyses are performed on the scRNAseq with appropriate ORA to group responses into GO terms. This is a new level of data for elephant immunology, but in terms of the relation to the vaccine a discussion on the rMVA alone should be included. Currently it is presented as EEHV-vaccine responses, but although cells were stimulated with the proteins some, or most, of the observations are likely more related to the live MVA vector than the selected antigens. Elephant studies are obviously not easy to perform and should be well justified, but studies (alternatively in a proxy species) with empty MVA vector, rMVA without w/o/w (and vice versa) would add another dimension to the understanding of the vaccine immunogenicity. This should at least be discussed.

Authors' Comment

We appreciate the reviewer's request to further discuss the immunostimulatory effects of rMVA and its potential contributions to the observed immune responses. In our manuscript, we refer collectively to the overall EEHV vaccine responses with the understanding that intrinsic and bystander effector properties of the MVA vector may contribute to and enhance (antigen-specific) immune responses. As noted earlier in relation to anti-vector responses, our heterologous prime-boost design is also intentionally employed to mitigate against significant immune responses to the vector. Nevertheless, specific contributions of the individual vaccine components would need to be elucidated further in future studies.

In our transcriptomic analyses, we first compared unstimulated whole blood samples obtained before and after vaccination (contrast A) and observed significant enrichment of CD8⁺ and CD4⁺ T cell gene sets alongside a downregulation of innate cell markers, notably those for CD14⁺ monocytes. Given that the respective samples were not antigen-stimulated an effect of the MVA vector cannot be excluded.

The analysis of antigen-stimulated differential gene expression and over-represented pathways (contrasts B and C) was relative to time-matched unstimulated samples at each timepoint, subtracting the background of the unstimulated samples that make up contrast A. The additional T and B cell-related responses observed upon *ex vivo* stimulation with recombinant EEHV proteins reflect an antigen-driven and -specific immune response not likely related to MVA/rMVA.

It is plausible and subject of recent discussions in immunology that MVA/rMVA may also have induced an innate immunity pre-activation, referred to as trained immunity¹. We measure immune responses after vaccination (contrast C) that match descriptions of trained immunity and further acknowledge this in the discussion accordingly.

Based on the above, we now also added the following part to the Discussion (p. 18, lines 503–513):

“While our study interprets post-vaccination immune signatures as EEHV antigen-specific responses elicited by the heterologous prime-boost regimen, it is important to acknowledge that we cannot delineate the contribution of the different vaccine components employed during the repeated heterologous prime-boost regime. Some effects may be attributable to the rMVA or other vaccine components, rather than exclusively to the vaccine antigens. For instance, it is possible that the rMVA through its MVA backbone has re-activated already existing, latency related T cell responses that underlie the results in contrast A where no antigen stimulation was applied. Similarly, we cannot draw a conclusion to which extent the MVA vector, as a live-attenuated virus, drove the putative trained immunity responses we see in the post vaccination ORA. The most prominent responses observed pre- and post-vaccination, however, were antigen-specific in contrasts B and D.”

The sentence thereafter was edited to:

*“Taken together, these results demonstrate that the vaccine induced an **overall** antigen-specific T cell-driven immune response as intended through our selection of T cell vaccine antigens and vaccine platforms.”*

References cited in this response:

1. Taks, E. J. M., Moorlag, S. J. C. F. M., Netea, M. G. & van der Meer, J. W. M. Shifting the immune memory paradigm: Trained immunity in viral infections. *Annu. Rev. Virol.* **9**, 469-489 (2022).

Minor comments:

ORA is not explained as an abbreviation (on p 11)

Authors’ Comment

The abbreviation for over-representation analysis (ORA) had already been introduced on p. 8, line 215. However, we agree that for clarity this should be spelled out in the subtitle, which is now on p.12, line 327, again and have reintroduced the abbreviation accordingly.

Page 14 second last paragraph: “revealed” should be replaced by “indicate”

Authors’ Comment

The word “revealed” has been replaced by “indicated” as suggested (now p. 15, lines 418 and 422, respectively).

Page 16 something is missing in front of “range countries”

Authors’ Comment

We have edited this sentence to be more specific as we are referring to EEHV vaccination trials that may be conducted within countries where Asian elephants are naturally found. To enhance clarity, the original sentence, which is now on p. 19, lines 525–527, was revised to:

“Thus, this pilot study has laid a foundation for the successful implementation of immunogenicity and efficacy vaccination trials in larger cohorts in Asian elephant range countries and may provide a roadmap for vaccination studies of other wildlife diseases.”

Page 17 first paragraph. Include a sentence of possible replication and shedding of the MVA vector in relation to safety profile.

Authors’ Comment

To follow the reviewer’s suggestion, we have incorporated the following statement where suggested (now p. 19, lines 536–542):

“MVA is a highly attenuated and replication-deficient vector classified as a Risk Group 1 agent, indicating minimal risk to both human health and the environment^{33, 34}. Although shedding data are rarely reported in clinical studies, available evidence suggests that MVA persists in the body for less than 48 hours, and detection of shed material in excreta is short-lived, consistent with rapid viral

clearance due to MVA's replication-deficient properties, which mitigates concerns about environmental release or transmission to untreated individuals⁷⁵."

References cited in the newly added manuscript text

33. Volz, A. & Sutter, G. Modified vaccinia virus Ankara: History, value in basic research, and current perspectives for vaccine development. *Adv. Virus Res.* **97**, 187-243 (2017).
34. Orlova, O. V., Glazkova, D. V., Bogoslovskaya, E. V., Shipulin, G. A. & Yudin, S. M. Development of modified vaccinia virus Ankara-based vaccines: Advantages and applications. *Vaccines* (Basel) **10**, 1516 (2022).
75. Verheust, C., Goossens, M., Pauwels, K. & Breyer, D. Biosafety aspects of modified vaccinia virus Ankara (MVA)-based vectors used for gene therapy or vaccination. *Vaccine* **30**, 2623-2632 (2012).

Page 23 last paragraph. With more than one sample for each elephant prior to vaccination, how is a t-test the right analysis?

Authors' Comment

We appreciate that we had phrased this ambiguously. We did not apply a standard t-test. but used the statistical framework implemented in the DESeq2 R package, which performs moderated hypothesis testing specifically tailored for count data from RNA sequencing. DESeq2 models the data using a negative binomial distribution, accounting for both biological and technical variability. The resulting test (while also referred to as moderated t-test within DESeq2) is rather a shrinkage-based Wald test, which incorporates empirical Bayes moderation to improve dispersion estimation across genes. This allows for more reliable inference, especially when sample sizes are small. This has been clarified in the manuscript Material and Method section (p. 27, lines 772–774).

Reviewer #3 (Remarks to the Author):

The authors present the novel findings and the first report of EEHV vaccine trial in adult Asian elephants. The sequence of the manuscript is easy to follow and well written with showing an alternative immunological diagnostic tool for EEHV vaccine research. Currently, there are several research groups have being investigated the EEHV vaccine trails in either adult or juvenile Asian elephants with no report yet. Despite the manuscript being interesting and suited to publication, there are some points that need to be addressed or discussed prior to publication.

Major point:

1. The major limitation of the present study is the number of animal samples used. Despite it being understood in case of experiments in such large, endangered animals like elephants, the previous history of animals showing the previous history of EEHV(s) exposure, immune status against EEHV major proteins, such as glycoproteins, should be addressed and mentioned in the Materials and Methods and result section.

Author's Comments

We appreciate the reviewer's considerate comments. We accept that our analysis was conducted using a limited number of animals, as the reviewer rightly recognises the limitations of such studies.

As the reviewer also correctly assumes, all elephants had been previously infected and historical information regarding their anamnestic status and previous infection was collated by Chester Zoo independently of this study in the context of routine herd health management and as part of a BSc Honours dissertation at the University of Liverpool¹ (now ⁴² in manuscript). Hence, we now explicitly mention that all three elephants had a history of EEHV detection and pre-existing humoral information in the Materials and Methods description of the study cohort on p. 23, lines 656–658.

As requested by another reviewer, we now include a Supplementary Fig. 2 presenting longitudinal antibody detection by western blot analysis. The results demonstrate low baseline levels of antibodies against the vaccine antigens, consistent with pre-existing EEHV immunity before immunisation and show an increase in antibody levels following vaccination.

In the Results section, we revised the paragraph discussing baseline *IFNG* responses to highlight that the observed recall responses are supported by historical infection and serological data (p. 6, lines 145–148). The sentence now reads:

“These recall responses before vaccination support the notion that all three adult elephants were latently infected with EEHV as anticipated for adult animals, which is corroborated by historical EEHV infection and serological data indicating humoral immunity to major viral glycoproteins⁴².”

References cited in this response:

1. Ward, E. *An investigation of elephant endotheliotropic herpesvirus (EEHV) shedding and viraemia in a herd of 10 Asian elephants (Elephas maximus) at a UK zoo in relation to births and deaths*. Intercalated Honours BSc dissertation in Veterinary Conservation Medicine, University of Liverpool (2023).

2. Although the authors aimed to investigate the cellular immune response using the EE2 and MCP internal proteins, brief comparison and discussion of this vaccine platform with other previous surface envelope glycoprotein platforms should also be mentioned in the discussion section.

Authors' Comment

We appreciate the reviewer's suggestion to provide a brief comparison of our vaccine platform with other, recent and ongoing approaches targeting surface envelope glycoproteins. To address this, we have inserted a new paragraph in the Discussion section immediately following the passage that describes the over-representation of pathways related to B cell activation (p. 17, lines 477–487).

This passage with the added text now reads as follows:

“While our vaccine targets effector T cell immune responses, whereby the Th cells might indirectly support a broad range of B cells upon infection and thus also promote a more rapid increase in neutralising antibodies, more direct approaches targeting EEHV-specific glycoproteins, particularly gB and the gH/gL complex, to elicit neutralising antibody responses have been undertaken in mice. The first two such studies tested the induction of gB-specific responses with rMVA and a protein subunit approach separately^{70, 71}. The most recent study uses a multivalent glycoprotein mRNA vaccine that demonstrated potential in a murine model⁷². Both neutralising antibody- and T cell-based strategies are valid avenues for EEHV vaccine development, and future vaccine development or optimisation efforts may benefit from integrating both strategies to enhance the breadth of immune responses and potentially extend cross-protection across EEHV genotypes.”

References cited in the newly added manuscript text:

70. Pursell, T., Spencer Clinton, J. L., Tan, J., Peng, R. & Ling, P. D. Modified vaccinia Ankara expressing EEHV1A glycoprotein B elicits humoral and cell-mediated immune responses in mice. *PLoS One* **17**, e0265424 (2022).
71. Spencer Clinton, J. L. et al. EEHV1A glycoprotein B subunit vaccine elicits humoral and cell-mediated immune responses in mice. *Vaccine* **40**, 5131–5140 (2022).
72. Watts, J. R. et al. Multi-antigen elephant endotheliotropic herpesvirus (EEHV) mRNA vaccine induces humoral and cell-mediated responses in mice. *Vaccines (Basel)* **12**, 1429 (2024).

Minor point:

1. Why the authors choose to investigate the heterologous vaccine platform for the first time in adult elephants instead of homologous vaccines. This hypothesis and theory should be mentioned in the introduction section.

Authors' Comment

We thank the reviewer for this suggestion. The primary reason for choosing a heterologous over a homologous vaccine regimen was that heterologous strategies have been demonstrated to induce broader, more efficacious, and more durable immune responses¹⁻³. We had mentioned this in the Introduction (p. 4, lines 101–104), but have now elaborated on further benefits that supported our choice.

We trust the expansion on p. 4, lines 104–114, now provides a clearer rationale for pursuing a heterologous vaccination strategy:

“Combining different vaccine modalities leverages their distinct immunological mechanisms to enhance the overall effect³⁰. Viral vectors are particularly effective at priming T cell-mediated immunity because they mimic natural viral infection and efficiently present intracellular antigens to

the host's immune system. In our vaccine design, this allowed us to exploit the capacity of MVA to robustly prime CD4+ and CD8+ T cells³¹, a key requirement for providing immunologically naïve elephant calves with an early and effective cellular immune response against primary EEHV infection. Complementing this, the use of a subunit boost vaccine formulated with an adjuvant directs the immune response toward the expansion of Th1 responses³², while using a distinct pathway of antigen delivery, processing and presentation. This strategy supports prolonged immune stimulation and facilitates the development of durable memory T cells.”

References cited in this response (excl. manuscript excerpt):

1. Kardani, K., Bolhassani, A. & Shahbazi, S. Prime-boost vaccine strategy against viral infections: Mechanisms and benefits. *Vaccine* **34**, 413–423 (2016).
2. Livieratos, A., Gogos, C., Thomas, I. & Akinosoglou, K. Vaccination strategies: Mixing paths versus matching tracks. *Vaccines* **13**, 308 (2025).
3. McCann, N., O'Connor, D., Lambe, T. & Pollard, A. J. Viral vector vaccines. *Curr. Opin. Immunol.* **77**, 102210 (2022).